# RELIABLE GENERATION OF EHR TIME SERIES VIA DIFFUSION MODELS

## ABSTRACT

Electronic Health Records (EHRs) are rich sources of patient-level data, including laboratory tests, medications, and diagnoses, offering valuable resources for medical data analysis. However, concerns about privacy often restrict access to EHRs, hindering downstream analysis. Researchers have explored various methods for generating privacy-preserving EHR data. In this study, we introduce a new method for generating diverse and realistic synthetic EHR time series data using Denoising Diffusion Probabilistic Models (DDPM). We conducted experiments on six datasets, comparing our proposed method with eight existing methods. Our results demonstrate that our approach significantly outperforms all existing methods in terms of data utility while requiring less training effort. Our approach also enhances downstream medical data analysis by providing diverse and realistic synthetic EHR data.

## 1 INTRODUCTION

The Electronic Health Record (EHR) is a digital version of the patient's medical history maintained by healthcare providers. It includes information such as demographic attributes, vital signals, and lab measurements that are sensitive in nature and important for clinical research. Researchers have been utilizing statistical and machine learning (ML) methods to analyze EHR for a variety of downstream tasks such as disease diagnosis, in-hospital mortality prediction, and disease phenotyping (Shickel et al., 2018; Goldstein et al., 2017). However, due to privacy concerns, EHR data is strictly regulated, and thus the availability of EHR data is often limited, creating barriers to the development of computational models in the field of healthcare. Widely used EHR de-identification methods to preserve patient information privacy are criticized for having high risks of re-identification of the individuals (Benitez & Malin, 2010).

Instead of applying privacy-preserving methods that can adversely affect EHR data utility (Janmey & Elkin, 2018), EHR synthetic data generation is one promising solution to protect patient privacy. Realistic synthetic data preserves crucial clinical information in real data while preventing patient information leakage (Yan et al., 2022; Yoon et al., 2023). Synthetic data also has the added benefit of providing a larger sample size for downstream analysis than de-identifying real samples (Gonzales et al., 2023). As a result, more research initiatives have begun to consider synthetic data sharing, such as the National COVID Cohort Collaborative supported by the U.S. National Institutes of Health and the Clinical Practice Research Datalink sponsored by the U.K. National Institute for Health and Care Research (Haendel et al., 2020; Herrett et al., 2015). With the advancement in ML techniques, applying generative models to synthesize high-fidelity EHR data is a popular research of interest (Yan et al., 2022). Recent advances in generative models have shown significant success in generating realistic high-dimensional data like images, audio, and texts (Gui et al., 2023; Yi et al., 2018), suggesting the potential for these models to handle EHR data with complex statistical characteristics.

Some representative work utilizing generative models for EHR data synthesis includes medGAN (Choi et al., 2017), medBGAN (Baowaly et al., 2019), and EHR-Safe (Yoon et al., 2023). However, most approaches to EHR data synthesis are GAN-based, and GANs are known for their difficulties in model training and deployments due to training instability and mode collapse (Saxena & Cao, 2021). Recently, diffusion probabilistic models have shown superb ability over GANs in generating high-fidelity image data (Ho et al., 2020; Nichol & Dhariwal, 2021a; Rombach et al., 2022). A few

studies thus propose to generate synthetic EHR data via diffusion models given their remarkable data generation performance (He et al., 2023; Yuan et al., 2023). However, most EHR data synthesis methods, either GAN-based or diffusion-based, focus on binary or categorical variables such as the International Classification of Diseases (ICD) codes. Additionally, there is limited prior work on generating EHR data with temporal information, and most state-of-the-art time series generative models are GAN-based. The sole study that employs diffusion models for EHR time series overlooks discrete time series in its modeling process (Kuo et al., 2023). It resorts to Gaussian diffusion for generating discrete sequences, treating them similarly to real-valued sequences but with further post-processing of the model output. These observations motivate us to bridge the gap by introducing a novel diffusion-based method to generate realistic EHR time series data with mixed variable types.

Specifically, we make the following contributions in this paper:

- We propose TIMEDIFF, a diffusion probabilistic model that uses a bidirectional recurrent neural network (BRNN) architecture for high-utility time series data generation.

- By combining multinomial and Gaussian diffusions, TIMEDIFF can simultaneously generate both real and discrete valued time series directly. To the best of our knowledge, TIMEDIFF is the first work in applying this mixed diffusion approach on EHR time series generation.

- We experimentally demonstrate that TIMEDIFF outperforms state-of-the-art methods for time series data generation by a big margin in terms of data utility. Additionally, our model requires less training effort compared to GANs.

- We further evaluate TIMEDIFF on potential applications in healthcare and show it can generate useful synthetic samples for ML model development while protecting patient privacy.

Our code is available upon request, and it will be made publicly available after acceptance.

## 2 RELATED WORK

**Time Series Generation:** Prior sequential generation methods using GANs rely primarily on binary adversarial feedback (Mogren, 2016; Esteban et al., 2017), and supervised sequence models mainly focus on tasks such as prediction (Dai & Le, 2015), forecasting (Lyu et al., 2018), and classification (Srivastava et al., 2016). TimeGAN (Yoon et al., 2019) was one of the first methods to preserve temporal dynamics in time series synthesis. The architecture comprises an embedding layer, recovery mechanism, generator, and discriminator, trained using both supervised and unsupervised losses. GT-GAN (Jeon et al., 2022) considers the generation of both regular and irregular time series data using a neural controlled differential equation (NCDE) encoder (Kidger et al., 2020) and GRU-ODE decoder (De Brouwer et al., 2019). This framework, combined with a continuous time flow processes (CTFPs) generator (Deng et al., 2021) and a GRU-ODE discriminator, outperformed existing methods in general-purpose time series generation. Recently, Biloš et al. (2023) proposed to generate time series data for forecasting and imputation using discrete or continuous stochastic process diffusion (DSPD/CSPD). Their proposed method views time series as discrete realizations of an underlying continuous function. Both DSPD and CSPD use either the Gaussian or Ornstein-Uhlenbec process to model noise and apply it to the entire time series. The learned distribution over continuous functions is then used to generate synthetic time series samples.

**Diffusion Models:** Diffusion models (Sohl-Dickstein et al., 2015) have been proposed and achieved excellent performance in the field of computer vision and natural language processing. Ho et al. (2020) proposed denoising diffusion probabilistic models (DDPM) that generate high-quality images by recovering from white latent noise. Gu et al. (2022) proposed a vector-quantized diffusion model on text-to-image synthesis with significant improvement over GANs regarding scene complexity and diversity of the generated images. Dhariwal & Nichol (2021) suggested that the diffusion models with optimized architecture outperform GANs on image synthesis tasks. Saharia et al. (2022) proposed a diffusion model, Imagen, incorporated with a language model for text-to-image synthesis with state-of-the-art results. Kotelnikov et al. (2022) introduced TabDDPM, an extension of DDPM for heterogeneous tabular data generation, outperforming GAN-based models. Das et al. (2023) proposed ChiroDiff, a diffusion model that considers temporal information and generates chirographic data. Besides advancements in practical applications, some recent developments in

theory for diffusion models demonstrate the effectiveness of this model class. Theoretical foundations explaining the empirical success of diffusion or score-based generative models have been established (Song & Ermon, 2019; 2020; Chen et al., 2022).

**EHR Data Generation:** There exists a considerable amount of prior work on generating EHR data. Choi et al. (2017) proposed medGAN that generates EHR discrete variables. Built upon medGAN, Baowaly et al. (2019) suggested two models, medBGAN and medWGAN, that synthesize EHR binary or discrete variables on International Classification of Diseases (ICD) codes. Yan et al. (2020) developed a GAN that can generate high-utility EHR with both discrete and continuous data. Biswal et al. (2021) proposed the EHR Variational Autoencoder that synthesizes sequences of EHR discrete variables (i.e., diagnosis, medications, and procedures). He et al. (2023) developed MedDiff, a diffusion model that generates user-conditioned EHR discrete variables. Yuan et al. (2023) created EHRDiff by utilizing the diffusion model to generate a collection of ICD diagnosis codes. Naseer et al. (2023) used continuous-time diffusion models to generate synthetic EHR tabular data. Ceritli et al. (2023) applied TabDDPM to synthesize tabular healthcare data.

However, most existing work focuses on discrete or tabular data generation. There is limited literature on EHR time series data generation, and this area of research has not yet received much attention (Koo & Kim, 2023). Back in 2017, RCGAN (Esteban et al., 2017) was created for generating multivariate medical time series data by employing RNNs as the generator and discriminator. Until recently, Yoon et al. (2023) proposed EHR-Safe that consists of a GAN and an encoder-decoder module. EHR-Safe can generate realistic time series and static variables in EHR with mixed data types. Li et al. (2023) developed EHR-M-GAN that generates mixed-type time series in EHR using separate encoders for each data type. Moreover, Kuo et al. (2023) suggested utilizing diffusion models to synthesize discrete and continuous EHR time series. However, their approach mainly relies on Gaussian diffusion and adopts a U-Net architecture (Ronneberger et al., 2015). The generation of discrete time series is achieved by taking argmax of softmax over real-valued one-hot representations. By contrast, our proposed method considers multinomial diffusion for discrete time series generation, allowing the generation of discrete variables directly.

## 3 DIFFUSION PROBABILISTIC MODELS

In this section, we explain diffusion models following the work of Sohl-Dickstein et al. (2015) and Ho et al. (2020). Diffusion models belong to a class of latent variable models formulated as $p_\theta(\boldsymbol{x}^{(0)}) = \int p_\theta(\boldsymbol{x}^{(0:T)}) \, d\boldsymbol{x}^{(1:T)}$, where $\boldsymbol{x}^{(0)}$ is a sample following the data distribution $q(\boldsymbol{x}^{(0)})$ and $\{\boldsymbol{x}^{(t)}\}_{t=1}^{T}$ are latents with the same dimensionality as $\boldsymbol{x}^{(0)}$.

The *forward process* is defined as a Markov chain that gradually adds Gaussian noise to $\boldsymbol{x}^{(0)}$ via a sequence of variances $\{\beta^{(t)}\}_{t=1}^{T}$:

$$q(\boldsymbol{x}^{(1:T)}|\boldsymbol{x}^{(0)}) = \prod_{t=1}^{T} q(\boldsymbol{x}^{(t)}|\boldsymbol{x}^{(t-1)}), \quad q(\boldsymbol{x}^{(t)}|\boldsymbol{x}^{(t-1)}) := \mathcal{N}(\boldsymbol{x}^{(t)}; \sqrt{1-\beta^{(t)}}\boldsymbol{x}^{(t-1)}, \beta^{(t)}\boldsymbol{I}). \quad (1)$$

The process successively converts data $\boldsymbol{x}^{(0)}$ to white latent noise $\boldsymbol{x}^{(T)}$. The noisy sample $\boldsymbol{x}^{(t)}$ can be obtained directly from the original sample $\boldsymbol{x}^{(0)}$ by sampling from $q(\boldsymbol{x}^{(t)}|\boldsymbol{x}^{(0)}) = \mathcal{N}(\boldsymbol{x}^{(t)}; \sqrt{\bar{\alpha}^{(t)}}\boldsymbol{x}^{(0)}, (1-\bar{\alpha}^{(t)})\boldsymbol{I})$, where $\alpha^{(t)} = 1 - \beta^{(t)}$ and $\bar{\alpha}^{(t)} = \prod_{i=1}^{t} \alpha^{(i)}$.

The *reverse process* is the joint distribution $p_\theta(\boldsymbol{x}^{(0:T)}) = p(\boldsymbol{x}^{(T)}) \prod_{t=1}^{T} p_\theta(\boldsymbol{x}^{(t-1)}|\boldsymbol{x}^{(t)})$, which is a Markov chain that starts from white latent noise and gradually denoises noisy samples to generate synthetic samples:

$$p_\theta(\boldsymbol{x}^{(t-1)}|\boldsymbol{x}^{(t)}) := \mathcal{N}(\boldsymbol{x}^{(t-1)}; \boldsymbol{\mu}_\theta(\boldsymbol{x}^{(t)}, t), \boldsymbol{\Sigma}_\theta(\boldsymbol{x}^{(t)}, t)), \quad p(\boldsymbol{x}^{(T)}) := \mathcal{N}(\boldsymbol{x}^{(T)}; \boldsymbol{0}, \boldsymbol{I}). \quad (2)$$

Under a specific parameterization described in Ho et al. (2020), the training objective can be expressed as follows:

$$\mathbb{E}_{\boldsymbol{x}^{(0)}, \boldsymbol{\epsilon}}\left[\frac{(\beta^{(t)})^2}{2(\sigma^{(t)})^2\alpha^{(t)}(1-\bar{\alpha}^{(t)})}\left\|\boldsymbol{\epsilon} - \boldsymbol{s}_\theta(\sqrt{\bar{\alpha}^{(t)}}\boldsymbol{x}^{(0)} + \sqrt{1-\bar{\alpha}^{(t)}}\boldsymbol{\epsilon}, t)\right\|^2\right] + C, \quad (3)$$

where $C$ is a constant that is not trainable. Empirically, a neural network $s_\theta$ is trained to approximate $\epsilon$. This $\epsilon$-prediction objective resembles denoising score matching, and the sampling procedure resembles Langevin dynamics using $s_\theta$ as an estimator of the gradient of the data distribution (Song & Ermon, 2019; 2020).

## 4 SYNTHETIC EHR TIME SERIES DATA WITH DIFFUSION MODELS

In this section, we discuss our methodology for generating realistic synthetic EHR time series data. We first introduce our notations. We consider the generation of both numerical (real-valued) and discrete time series in our framework, as both are present in EHR. Specifically, let $\mathcal{D}$ denote our EHR time series dataset. Each patient in $\mathcal{D}$ has numerical and discrete multivariate time series $\boldsymbol{X} \in \mathbb{R}^{P_r \times L}$ and $\boldsymbol{C} \in \mathbb{Z}^{P_d \times L}$, respectively. $L$ is the number of time steps, and $P_r$ and $P_d$ are the number of variables for numerical and discrete data types.

### 4.1 DIFFUSION PROCESS ON EHR TIME SERIES

To generate both numerical and discrete time series, we consider a "mixed sequence diffusion" approach by adding Gaussian and multinomial noises. For numerical time series, we perform Gaussian diffusion by adding independent Gaussian noise similar to DDPM. The forward process is thus defined as:

$$q\big(\boldsymbol{X}^{(1:T)}|\boldsymbol{X}^{(0)}\big) = \prod_{t=1}^{T}\prod_{l=1}^{L} q\big(\boldsymbol{X}_{\cdot,l}^{(t)}|\boldsymbol{X}_{\cdot,l}^{(t-1)}\big), \tag{4}$$

where $q(\boldsymbol{X}_{\cdot,l}^{(t)}|\boldsymbol{X}_{\cdot,l}^{(t-1)}) = \mathcal{N}(\boldsymbol{X}_{\cdot,l}^{(t)}; \sqrt{1-\beta^{(t)}}\boldsymbol{X}_{\cdot,l}^{(t-1)}, \beta^{(t)}\boldsymbol{I})$ and $\boldsymbol{X}_{\cdot,l}$ is the $l^{\text{th}}$ observation of the numerical time series. In a similar fashion as Equation (2), we define the reverse process for numerical features as $p_\theta(\boldsymbol{X}^{(0:T)}) = p(\boldsymbol{X}^{(T)}) \prod_{t=1}^{T} p_\theta\big(\boldsymbol{X}^{(t-1)}|\boldsymbol{X}^{(t)}\big)$, where

$$p_\theta\big(\boldsymbol{X}^{(t-1)}|\boldsymbol{X}^{(t)}\big) := \mathcal{N}\big(\boldsymbol{X}^{(t-1)}; \boldsymbol{\mu}_\theta(\boldsymbol{X}^{(t)}, t), \tilde{\beta}^{(t)}\boldsymbol{I}\big),$$

$$\boldsymbol{\mu}_\theta(\boldsymbol{X}^{(t)}, t) = \frac{1}{\sqrt{\alpha^{(t)}}}\left(\boldsymbol{X}^{(t)} - \frac{\beta^{(t)}}{\sqrt{1-\bar{\alpha}^{(t)}}}\boldsymbol{s}_\theta(\boldsymbol{X}^{(t)}, t)\right), \quad \tilde{\beta}^{(t)} = \frac{1-\bar{\alpha}^{(t-1)}}{1-\bar{\alpha}^{(t)}}\beta^{(t)}. \tag{5}$$

In order to model discrete-valued time series, we use multinomial diffusion (Hoogeboom et al., 2021). The forward process is defined as:

$$q\big(\tilde{\boldsymbol{C}}^{(1:T)}|\tilde{\boldsymbol{C}}^{(0)}\big) = \prod_{t=1}^{T}\prod_{p=1}^{P_d}\prod_{l=1}^{L} q\big(\tilde{\boldsymbol{C}}_{p,l}^{(t)}|\tilde{\boldsymbol{C}}_{p,l}^{(t-1)}\big), \tag{6}$$

$$q\big(\tilde{\boldsymbol{C}}_{p,l}^{(t)}|\tilde{\boldsymbol{C}}_{p,l}^{(t-1)}\big) := \mathcal{C}\big(\tilde{\boldsymbol{C}}_{p,l}^{(t)}; (1-\beta^{(t)})\tilde{\boldsymbol{C}}_{p,l}^{(t-1)} + \beta^{(t)}/K\big), \tag{7}$$

where $\mathcal{C}$ is a categorical distribution, $\tilde{\boldsymbol{C}}_{p,l}^{(0)} \in \{0,1\}^K$ is a one-hot encoded representation of $C_{p,l}$[1], and the addition and subtraction between scalars and vectors are performed element-wise. The forward process posterior distribution is defined as follows:

$$q\big(\tilde{\boldsymbol{C}}_{p,l}^{(t-1)}|\tilde{\boldsymbol{C}}_{p,l}^{(t)}, \tilde{\boldsymbol{C}}_{p,l}^{(0)}\big) := \mathcal{C}\left(\tilde{\boldsymbol{C}}_{p,l}^{(t-1)}; \boldsymbol{\phi}/\sum_{k=1}^{K}\phi_k\right), \tag{8}$$

$$\boldsymbol{\phi} = \big(\alpha^{(t)}\tilde{\boldsymbol{C}}_{p,l}^{(t)} + (1-\alpha^{(t)})/K\big) \odot \big(\bar{\alpha}^{(t-1)}\tilde{\boldsymbol{C}}_{p,l}^{(0)} + (1-\bar{\alpha}^{(t-1)})/K\big). \tag{9}$$

The reverse process $p_\theta(\tilde{\boldsymbol{C}}_{p,l}^{(t-1)}|\tilde{\boldsymbol{C}}_{p,l}^{(t)})$ is parameterized as $q(\tilde{\boldsymbol{C}}_{p,l}^{(t-1)}|\tilde{\boldsymbol{C}}_{p,l}^{(t)}, \boldsymbol{s}_\theta(\tilde{\boldsymbol{C}}_{p,l}^{(t)}, t))$. We train our neural network, $\boldsymbol{s}_\theta$, using both Gaussian and multinomial diffusion processes:

$$\mathcal{L}_{\mathcal{N}}(\theta) := \mathbb{E}_{\boldsymbol{X}^{(0)}, \epsilon, t}\left[\left\|\epsilon - \boldsymbol{s}_\theta\big(\sqrt{\bar{\alpha}^{(t)}}\boldsymbol{X}^{(0)} + \sqrt{1-\bar{\alpha}^{(t)}}\epsilon, t\big)\right\|^2\right], \tag{10}$$

$$\mathcal{L}_{\mathcal{C}}(\theta) := \mathbb{E}_{p,l}\left[\sum_{t=2}^{T} D_{\text{KL}}\Big(q(\tilde{\boldsymbol{C}}_{p,l}^{(t-1)}|\tilde{\boldsymbol{C}}_{p,l}^{(t)}, \tilde{\boldsymbol{C}}_{p,l}^{(0)}) \,\Big\|\, p_\theta(\tilde{\boldsymbol{C}}_{p,l}^{(t-1)}|\tilde{\boldsymbol{C}}_{p,l}^{(t)})\Big)\right], \tag{11}$$

---

[1]We perform one-hot encoding on the discrete time series across the feature dimension. For example, if our time series is $\{0, 1, 2\}$, its one-hot representation becomes $\{[1,0,0]^\top, [0,1,0]^\top, [0,0,1]^\top\}$.

where $\mathcal{L}_{\mathcal{N}}$ and $\mathcal{L}_{\mathcal{C}}$ are the losses for numerical and discrete multivariate time series, respectively. The training of the neural network is performed by minimizing the following loss:

$$\mathcal{L}_{\text{train}}(\theta) = \lambda \mathcal{L}_{\mathcal{C}}(\theta) + \mathcal{L}_{\mathcal{N}}(\theta), \tag{12}$$

where $\lambda$ is a hyperparameter for creating a balance between the two losses. We investigate the effects of $\lambda$ in Appendix B.3.

## 4.2 MISSING VALUE REPRESENTATION

In medical applications, missing data and variable measurement times play a crucial role as they could provide additional information and indicate a patient's health status (Zhou et al., 2023). We thus derive a missing indicator mask $M \in \{0,1\}^{P_r \times L}$ [2] for each $X \in \mathcal{D}$ [3]:

$$M_{p,l} = \begin{cases} 0, & \text{if } X_{p,l} \text{ is present;} \\ 1, & \text{if } X_{p,l} \text{ is missing.} \end{cases} \tag{13}$$

Then $M$ encodes the measurement time points of $X$. If $X$ contains missing values, we impute them in the initial value of the forward process, i.e., $X^{(0)}$, using the corresponding sample mean. Nevertheless, $M$ retains the information regarding the positions of missing values. Our method generates both discrete and numerical time series, allowing us to represent and generate $M$ as a discrete time series seamlessly.

## 4.3 TIMEDIFF ARCHITECTURE

In this section, we describe our architecture for the diffusion model. A commonly used architecture in DDPM is U-Net (Ronneberger et al., 2015). However, most U-Net-based models are tailored to image generation tasks, requiring the neural network to process pixel-based data rather than sequential information (Song et al., 2020; Ho et al., 2020; Rombach et al., 2022). Even its one-dimensional variant, 1D-U-Net, comes with limitations such as restriction on the input sequence length (which must be a multiple of U-Net multipliers) and a tendency to lose temporal dynamics information during down-sampling. On the other hand, TabDDPM (Kotelnikov et al., 2022) proposed a mixed diffusion approach for tabular data generation but relied on a multilayer perceptron architecture, making it improper for multivariate time series generation.

To address this challenge of handling EHR time series, we need an architecture capable of encoding sequential information while being flexible to the input sequence length. The time-conditional bidirectional RNN (BRNN) or neural controlled differential equation (NCDE) (Kidger et al., 2020) can be possible options. After careful evaluation, we found BRNN without attention mechanism offers superior computational efficiency and have chosen it as the neural backbone $s_\theta$ for all of our experiments. A more detailed discussion of NCDE is provided in Appendix A.3.1.

**Diffusion Step Embedding:** To inform the model about current diffusion time step $t$, we use sinusoidal positional embedding (Vaswani et al., 2017). The embedding vector output from the embedding layer then goes through two fully connected (FC) layers with GeLU activation in between (Hendrycks & Gimpel, 2016). The embedding vector is then fed to a SiLU activation (Hendrycks & Gimpel, 2016) and another FC layer. The purpose of this additional FC layer is to adjust the dimensionality of the embedding vector to match the stacked hidden states from BRNN. Specifically, we set the dimensionality of the output to be two times the size of the hidden dimension from BRNN. We denote the transformed embedding vector as $t_{\text{embed}}$. This vector is then split into two vectors, each with half of the current size, namely $t_{\text{embed\_scale}}$ and $t_{\text{embed\_shift}}$. Both vectors share the same dimensionality as BRNN's hidden states and serve to inform the network about the current diffusion time step.

**Time-conditional BRNN:** In practice, BRNN can be implemented with either LSTM or GRU units. To condition BRNN on time, we follow these steps. We first obtain noisy samples from Gaussian (for numerical data) and multinomial (for discrete data) diffusion. The two samples are concatenated and fed to our BRNN, which returns a sequence of hidden states $\{h_l\}_{l=1}^L$ that stores the temporal

---

[2]Or alternatively, $M \in \{0,1\}^{P_d \times L}$ if the time series is discrete.

[3]For simplicity in writing, we refer to $X$ only, but this procedure can also be applied on $C$.

dynamics information about the time series. To stabilize learning and enable proper utilization of $t_{\text{embed}}$, we apply layernorm (Ba et al., 2016) on $\{h_l\}_{l=1}^L$. The normalized sequence of hidden states, $\{\tilde{h}_l\}_{l=1}^L$, is then scaled and shifted using $\{\tilde{h}_l \odot (t_{\text{embed\_scale}} + \mathbf{1}) + t_{\text{embed\_shift}}\}_{l=1}^L$. These scaled hidden states contain information about the current diffusion step $t$, which is then passed through an FC layer to produce the final output. The output contains predictions for both multinomial and Gaussian diffusions, which are extracted correspondingly and used to calculate $\mathcal{L}_{\text{train}}$ in Equation (12).

## 5 EXPERIMENTS

**Datasets:** We use four publicly available EHR datasets to evaluate TIMEDIFF: Medical Information Mart for Intensive Care III and IV (MIMIC III/IV) (Johnson et al., 2016; 2023), the eICU Collaborative Research Database (eICU) (Pollard et al., 2018), and high time resolution ICU dataset (HiRID) (Hyland et al., 2020). In order to evaluate TIMEDIFF with state-of-the-art methods for time series generation on non-EHR datasets, we include Stocks and Energy datasets used in studies that proposed TimeGAN (Yoon et al., 2019) and GT-GAN (Jeon et al., 2022).

**Baselines:** We compare TIMEDIFF with eight methods: EHR-M-GAN (Li et al., 2023), GT-GAN (Jeon et al., 2022), TimeGAN (Yoon et al., 2019), RCGAN (Esteban et al., 2017), C-RNN-GAN (Mogren, 2016), RNNs trained with teacher forcing (T-Forcing) (Graves, 2013; Sutskever et al., 2011) and professor forcing (P-Forcing) (Lamb et al., 2016), and discrete or continuous stochastic process diffusion (DSPD/CSPD) with Gaussian (GP) or Ornstein-Uhlenbeck (OU) processes (Biloš et al., 2023).

**Metrics:** (1-2) *Discriminative and Predictive Scores:* A GRU-based discriminator is trained to distinguish between real and synthetic samples. The discriminative score is $|0.5 - \text{Accuracy}|$. For the predictive score, a GRU-based predictor is trained on synthetic samples and evaluated on real samples for next-step vector prediction based on mean absolute error over each sequence.

(3) *Train on Synthetic, Test on Real (TSTR):* We train ML models entirely on synthetic data and evaluate them on real test data based on the area under the receiver operating characteristic curve (AUC) for in-hospital mortality prediction. We compare the TSTR score to the Train on Real, Test on Real (TRTR) score, which is the AUC obtained from the model trained on real training data and evaluated on real test data.

(4) *Train on Synthetic and Real, Test on Real (TSRTR):* Similar to TSTR, we train ML models and evaluate them on real test data using AUC. We fix the size of real training data to 2000 and add more synthetic samples to train ML models. This metric evaluates the impact on ML models when their training data includes an increasing amount of synthetic data. It also simulates the practical scenario where practitioners use synthetic samples to increase the sample size of the training data for model development.

(5) *t-SNE:* We flatten the feature dimension and use t-SNE dimension reduction visualization (Van der Maaten & Hinton, 2008) on synthetic, real training, and real testing samples. This qualitative metric measures the similarity of synthetic and real samples in two-dimensional space.

(6) *Nearest Neighbor Adversarial Accuracy Risk (NNAA):* This score measures the degree to which a generative model overfits the real training data, a factor that could raise privacy-related concerns (Yale et al., 2020). It is the difference between two discriminative accuracies, $AA_{\text{test}}$ and $AA_{\text{train}}$.

(7) *Membership Inference Risk (MIR):* An F1 score is calculated based on whether an adversary can correctly identify the membership of a synthetic data sample (Liu et al., 2019).

For all the experiments, we split each dataset into training and testing sets and use the training set to develop generative models. The synthetic samples obtained from trained generative models are then used for evaluation. We repeat each experiment over 10 times and report the mean and standard deviation of each quantitative metric. Further details for our experiments and evaluation metrics are discussed in Appendix A.

Table 1: Comparison of predictive and discriminative scores between TIMEDIFF and the baselines.

| Metric | Method | Stocks | Energy | MIMIC-III | MIMIC-IV | HiRID | eICU |
|---|---|---|---|---|---|---|---|
| | **TIMEDIFF** | **.048±.028** | **.088±.018** | **.028±.023** | **.030±.022** | **.333±.056** | **.015±.007** |
| | EHR-M-GAN | .483±.027 | .497±.006 | .499±.002 | .499±.001 | .496±.003 | .488±.022 |
| | DSPD-GP | .081±.034 | .416±.016 | .491±.002 | .478±.020 | .489±.004 | .327±.020 |
| | DSPD-OU | .098±.030 | .290±.010 | .456±.014 | .444±.037 | .481±.007 | .367±.018 |
| | CSPD-GP | .313±.061 | .392±.007 | .498±.001 | .488±.010 | .485±.007 | .489±.010 |
| Discriminative | CSPD-OU | .283±.039 | .384±.012 | .494±.002 | .479±.005 | .489±.004 | .479±.017 |
| Score | GT-GAN | .077±.031 | .221±.068 | .488±.026 | .472±.014 | .455±.015 | .448±.043 |
| ($\downarrow$) | TimeGAN | .102±.021 | .236±.012 | .473±.019 | .452±.027 | .498±.002 | .434±.061 |
| | RCGAN | .196±.027 | .336±.017 | .498±.001 | .490±.003 | .499±.001 | .490±.023 |
| | C-RNN-GAN | .399±.028 | .499±.001 | .500±.000 | .499±.000 | .499±.001 | .493±.010 |
| | T-Forcing | .226±.035 | .483±.004 | .499±.001 | .497±.002 | .480±.010 | .479±.011 |
| | P-Forcing | .257±.026 | .412±.006 | .494±.006 | .498±.002 | .494±.004 | .367±.047 |
| | *Real Data* | *.019±.016* | *.016±.006* | *.012±.006* | *.014±.011* | *.014±.015* | *.004±.003* |
| | **TIMEDIFF** | **.037±.000** | **.251±.000** | **.469±.003** | **.432±.002** | **.292±.018** | **.309±.019** |
| | EHR-M-GAN | .120±.047 | .254±.001 | .861±.072 | .880±.079 | .624±.028 | .913±.179 |
| | DSPD-GP | .038±.000 | .260±.001 | .509±.014 | .586±.026 | .404±.013 | .320±.018 |
| | DSPD-OU | .039±.000 | .252±.000 | .497±.006 | .474±.023 | .397±.024 | .317±.023 |
| | CSPD-GP | .041±.000 | .257±.001 | 1.083±.002 | .496±.034 | .341±.029 | .624±.066 |
| Predictive | CSPD-OU | .044±.000 | .253±.000 | .566±.006 | .516±.051 | .439±.010 | .382±.026 |
| Score | GT-GAN | .040±.000 | .312±.002 | .584±.010 | .517±.016 | .386±.033 | .487±.033 |
| ($\downarrow$) | TimeGAN | .038±.001 | .273±.004 | .727±.010 | .548±.022 | .729±.039 | .367±.025 |
| | RCGAN | .040±.001 | .292±.005 | .837±.040 | .700±.014 | .675±.074 | .890±.017 |
| | C-RNN-GAN | .038±.000 | .483±.005 | .933±.046 | .811±.048 | .727±.082 | .769±.045 |
| | T-Forcing | .038±.001 | .315±.005 | .840±.013 | .641±.017 | .364±.018 | .547±.069 |
| | P-Forcing | .043±.001 | .303±.006 | .683±.031 | .557±.030 | .445±.018 | .345±.021 |
| | *Real Data* | *.036±.001* | *.250±.003* | *.467±.005* | *.433±.001* | *.267±.012* | *.304±.017* |

TIMEDIFF    EHR-M-GAN    DSPD-GP    GT-GAN    TimeGAN    RCGAN

Figure 1: t-SNE for eICU (1st row) and MIMIC-IV (2rd row). Synthetic samples in **blue**, real training samples in **red**, and real testing samples in **orange**.

## 5.1 RESULTS

**Predictive and Discriminative Scores:** As presented in Table 1, we observe that TIMEDIFF consistently achieves the lowest discriminative and predictive scores across six datasets compared to all baselines. TIMEDIFF achieves significantly lower discriminative scores and close-to-real predictive scores on all four EHR datasets. For instance, TIMEDIFF yields a 95.4% lower mean discriminative score compared to DSPD-GP and obtains a 1.6% higher mean predictive score than real testing data on the eICU dataset. For non-EHR datasets, TIMEDIFF achieves a 37.7% lower and a 60.2% lower mean discriminative scores on the Stocks and Energy datasets than GT-GAN while having similar mean predictive scores as using real testing data.

**t-SNE:** As shown in Figure 1, the synthetic samples produced by TIMEDIFF demonstrate remarkable overlap with both real training and testing data, indicating their high fidelity. We have the same observations across all datasets, and the rest of the visualizations are in Appendix B.1.

**Runtime:** We compare the number of hours to train TIMEDIFF with EHR-M-GAN, TimeGAN, and GT-GAN. We used Intel Xeon Gold 6226 Processor and Nvidia GeForce 2080 RTX Ti for runtime comparison of all models. As indicated by Table 2, TIMEDIFF can produce high-fidelity synthetic samples with less training time compared to GAN-based approaches.

**In-hospital Mortality Prediction:** As introduced in Section 1, in-hospital mortality prediction is one of the most important downstream tasks utilizing EHR data (Sadeghi et al., 2018; Sheikhalishahi

et al., 2019). To evaluate the utility of the generated EHR time series samples using TIMEDIFF, we perform in-hospital mortality prediction using six ML algorithms: XGBoost (XGB) (Chen & Guestrin, 2016), Random Forest (RF) (Breiman, 2001), AdaBoost (AB) (Freund & Schapire, 1997), and $\ell_1$ and $\ell_2$ regularized Logistic Regression (LR L1/L2) (Friedman et al., 2010). The prediction models are trained using synthetic samples from TIMEDIFF and assessed on real testing data.

As indicated in Figure 2, we observe that models trained using pure synthetic samples have similar AUCs compared to those trained on the real training data. Furthermore, to simulate the practical scenario where synthetic samples are used for data augmentation, we calculate the TSRTR scores for each ML model. We observe that most ML models achieve better performances as more synthetic samples are added. This observation is also consistent with our previous findings, demonstrating the high fidelity of our synthetic data.

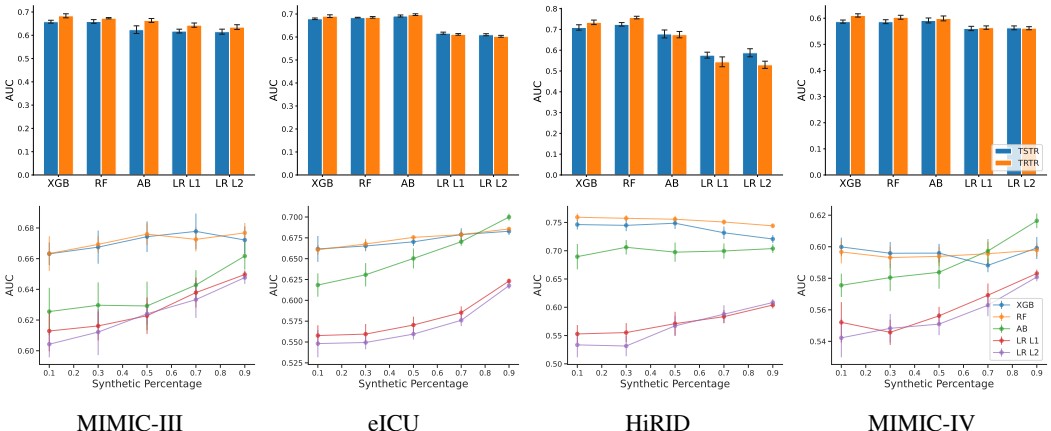

Figure 2: (Top) comparison of TSTR with TRTR scores; (Bottom) TSRTR score.

**NNAA and MIR:** As indicated in Table 4, we observe that TIMEDIFF consistently scores around 0.5 for both $AA_\text{test}$ and $AA_\text{train}$ while having low NNAA and MIR scores. This suggests that TIMEDIFF produces high-fidelity synthetic samples and does not overfit its train-

Table 2: Runtime comparisons (hours).

| Dataset | **TIMEDIFF** | EHR-M-GAN | TimeGAN | GT-GAN |
|---|---|---|---|---|
| MIMIC-III | **2.7** | 18.9 | 10.8 | 21.8 |
| MIMIC-IV | **2.7** | 28.8 | 29.5 | 47.3 |
| HiRID | **2.5** | 29.7 | 46.2 | 58.3 |
| eICU | **8.7** | 87.1 | 110 | 59.1 |

ing data. By contrast, although still mostly having low NNAA and MIR scores, all the baselines have higher $AA_\text{test}$ and $AA_\text{train}$. The full results are presented in Appendix B.4.

**Ablation Study:** We further investigate whether performing multinomial diffusion on missing indicators for discrete sequence generation is useful. We compare our method with Gaussian diffusion on the missing indicators, and these post-processing methods are applied to transform real-valued model predictions into discrete sequences: (1) direct rounding; (2) argmax on the softmax of real-valued, one-hot encoded representations[4]. We compare these methods on MIMIC-III/IV and eICU. HiRID is excluded from this ablation study since it is complete and does not contain missing values. Table 3 shows that the synthetic data has much higher utility when multinomial diffusion is adopted.

Table 3: Ablation study on generating missing indicators using multinomial diffusion.

| Metric | Method | MIMIC-III | MIMIC-IV | eICU |
|---|---|---|---|---|
| Discriminative Score ($\downarrow$) | with Gaussian and rounding | .355±.020 | .121±.025 | .030±.018 |
| | with Gaussian and softmax | .088±.023 | .155 ±.032 | .042±.045 |
| | with multinomial | **.028±.023** | **.030±.022** | **.015±.007** |
| Predictive Score ($\downarrow$) | with Gaussian and rounding | .486±.005 | .433±.003 | .312±.031 |
| | with Gaussian and softmax | .472±.004 | .434±.002 | .320±.035 |
| | with multinomial | **.469±.003** | **.432±.002** | **.309±.019** |

---

[4]The synthetic one-hot encoding is not discrete since Gaussian diffusion is used. This method is also adopted by Kuo et al. (2023) for the generation of discrete time series with diffusion models.

Table 4: Privacy score evaluations.

| Metric | Method | MIMIC-III | MIMIC-IV | HiRID | eICU |
|---|---|---|---|---|---|
| $AA_{\text{test}}$ ($\sim$0.5) | **TIMEDIFF** | **.574**±**.002** | **.517**±**.002** | **.531**±**.003** | **.537**±**.001** |
| | EHR-M-GAN | .998±.000 | 1.000±.000 | 1.000±.000 | .977±.000 |
| | DSPD-GP | .974±.001 | .621±.002 | .838±.004 | .888±.000 |
| | DSPD-OU | .927±.000 | .804±.003 | .886±.001 | .971±.000 |
| | CSPD-GP | .944±.001 | .623±.002 | .958±.002 | .851±.001 |
| | CSPD-OU | .967±.001 | .875±.002 | .947±.001 | .982±.000 |
| | GT-GAN | .995±.000 | .910±.001 | .990±.001 | .981±.000 |
| | TimeGAN | .997±.000 | .974±.001 | .643±.003 | 1.000±.000 |
| | RCGAN | .983±.001 | .999±.000 | 1.000±.000 | 1.000±.000 |
| | *Real Data* | *.552±.002* | *.497±.002* | *.511±.006* | *.501±.002* |
| $AA_{\text{train}}$ ($\sim$0.5) | **TIMEDIFF** | **.573**±**.002** | **.515**±**.002** | **.531**±**.002** | **.531**±**.002** |
| | EHR-M-GAN | .999±.000 | 1.000±.000 | 1.000±.000 | .965±.002 |
| | DSPD-GP | .968±.002 | .620±.003 | .851±.005 | .888±.001 |
| | DSPD-OU | .928±.001 | .788±.003 | .876±.002 | .971±.000 |
| | CSPD-GP | .940±.002 | .629±.005 | .965±.004 | .852±.001 |
| | CSPD-OU | .966±.001 | .880±.003 | .945±.002 | .983±.000 |
| | GT-GAN | .995±.001 | .907±.002 | .989±.001 | .981±.000 |
| | TimeGAN | .997±.000 | .969±.003 | .651±.004 | 1.000±.000 |
| | RCGAN | .984±.001 | .999±.000 | 1.000±.000 | 1.000±.000 |
| | *Real Data* | *.286±.003* | *.268±.004* | *.327±.006* | *.266±.002* |
| NNAA ($\downarrow$) | **TIMEDIFF** | .002±.002 | .002±.002 | .004±.003 | .006±.002 |
| | EHR-M-GAN | .000±.000 | .000±.000 | .000±.000 | .012±.003 |
| | DSPD-GP | .005±.003 | .003±.003 | .013±.007 | .001±.001 |
| | DSPD-OU | .001±.001 | .016±.004 | .010±.002 | .000±.000 |
| | CSPD-GP | .004±.002 | .007±.005 | .008±.004 | .001±.001 |
| | CSPD-OU | .001±.001 | .005±.003 | .002±.001 | .001±.001 |
| | GT-GAN | .001±.000 | .004±.002 | .001±.001 | .000±.000 |
| | TimeGAN | .000±.000 | .005±.003 | .008±.004 | .000±.000 |
| | RCGAN | .001±.000 | .000±.000 | .000±.000 | .000±.000 |
| | *Real Data* | *.267±.004* | *.229±.003* | *.184±.006* | *.235±.003* |
| MIR ($\downarrow$) | **TIMEDIFF** | .191±.008 | .232±.048 | .236±.179 | .227±.021 |
| | EHR-M-GAN | .025±.007 | .435±.031 | .459±.161 | .049±.006 |
| | DSPD-GP | .032±.021 | .050±.009 | .106 ±.064 | .000±.000 |
| | DSPD-OU | .060±.032 | .007±.006 | .005±.005 | .000±.000 |
| | CSPD-GP | .060±.028 | .034±.017 | .004±.004 | .000±.000 |
| | CSPD-OU | .066±.046 | .016±.020 | .005±.003 | .000±.000 |
| | GT-GAN | .005±.002 | .046±.013 | .109±.057 | .000±.000 |
| | TimeGAN | .010±.002 | .173±.020 | .624±.006 | .000±.000 |
| | RCGAN | .013±.002 | .277±.049 | .063±.013 | .000±.000 |
| | *Real Data* | *.948±.000* | *.929±.005* | *.737±.011* | *.927±.001* |

## 6 CONCLUSIONS, FUTURE WORK, AND LIMITATIONS

We propose TIMEDIFF for EHR time series generation by using mixed sequence diffusion and demonstrate its superior performance compared with all state-of-the-art time series generation methods in terms of data utility. We also demonstrate that TIMEDIFF can facilitate downstream analysis in healthcare while protecting patient privacy. However, it is important to acknowledge the limitations of our study. While our results suggest that TIMEDIFF offers some degree of patient privacy protection, it should not be seen as a replacement for official audits, which may still be necessary prior to data sharing. It is also interesting to investigate TIMEDIFF within established privacy frameworks, e.g., differential privacy. Additionally, to provide better interpretability and explainability of TIMEDIFF, subgroup analysis and theoretical analysis are to be developed. Lastly, it would also be meaningful to investigate the modeling of highly sparse and irregular temporal data, such as lab tests and medications. We leave the above potential improvements of TIMEDIFF for future work.

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

# Supplementary Text to "Reliable Generation of EHR Time Series via Diffusion Models"

## Table of Contents

# A EXPERIMENT DETAILS

## A.1 DATASETS

In this section, we provide further information on the datasets used in this study and the corresponding data processing procedures. Unless specified otherwise, all datasets are normalized by min-max scaling for model training, and the minimums and maximums are calculated feature-wise, i.e., we normalize each feature by its corresponding sample minimum and maximum, and this procedure is applied across all the features. For all EHR datasets, we extract the in-hospital mortality status as our class labels for TSTR and TSRTR evaluations.

Table 5: Dataset statistics.

| Dataset | Sample Size | Number of Features | Sequence Length | Missing (%) | Mortality Rate (%) |
|---|---|---|---|---|---|
| Stocks | 3,773 | 6 | 24 | 0 | — |
| Energy | 19,711 | 28 | 24 | 0 | — |
| MIMIC-III | 26,150 | 15 | 25 | 17.9 | 7.98 |
| MIMIC-IV | 21,593 | 11 | 72 | 7.9 | 23.67 |
| HiRID | 6,709 | 8 | 100 | 0 | 16.83 |
| eICU | 62,453 | 9 | 276 | 10.5 | 10.63 |

### A.1.1 STOCKS & ENERGY

As mentioned earlier in Section 5, we use the Stocks and Energy datasets for a fair comparison between TIMEDIFF and the existing GAN-based time-series generation methods. Both datasets can be downloaded from TimeGAN's official repository.

**Stocks:** The Stocks dataset contains daily Google stock data recorded between 2004 and 2019. It contains features such as volume, high, low, opening, closing, and adjusted closing prices. Each data point represents the value of those six features on a single day. The dataset is available online and can be accessed from the historical Google stock price on Yahoo.

**Energy:** The Energy dataset is from the UCI Appliances energy prediction data. It contains multivariate continuous-valued time-series and has high-dimensional, correlated, and periodic features. This dataset can be obtained from UCI machine learning repository.

To prepare both datasets for training and ensure consistency with previous approaches for a fair comparison, we use the same procedure as TimeGAN. We then apply training and testing splits for both datasets. For the Stocks dataset, we use 80% for training and 20% for testing. For the Energy dataset, we use 75% for training and 25% for testing.

### A.1.2 MIMIC-III

The Medical Information Mart for Intensive Care-III (MIMIC-III) is a single-center database consisting of a large collection of EHR data for patients admitted to critical care units at Beth Israel Deaconess Medical Center between 2001 and 2012. The dataset contains information such as demographics, lab results, vital measurements, procedures, caregiver notes, and patient outcomes. It contains data for 38,597 distinct adult patients and 49,785 hospital admissions.

**Variable Selection:** In our study, we use the following vital sign measurements from MIMIC-III: heart rate (beats per minute), systolic blood pressure (mm Hg), diastolic blood pressure (mm Hg), mean blood pressure (mm Hg), respiratory rate (breaths per minute), body temperature (Celsius), and oxygen saturation (%). To ensure consistency and reproducibility, we adopt the scripts in official MIMIC-III repository for data pre-processing that selects the aforementioned features based on *itemid* and filters potential outliers[5]. We then extract records of the selected variables within the first 24 hours of a patient's unit stay at one-hour intervals, where the initial measurement is treated as time step 0. This procedure gives us a multivariate time series of length 25 for each patient.

**Cohort Selection:** We select our MIMIC-III study cohort by applying the outlier filter criteria adopted by the official MIMIC-III repository. The filtering rules can be accessed here. We select

---

[5]For sake of reproducibility, the thresholds for the outliers are defined by the official repository.

patients based on the unit stay level using *icustay_id*. We only include patients who have spent at least 24 hours in their ICU stay.

We use 80% of the dataset for training and 20% for testing while ensuring a similar class ratio between the splits.

### A.1.3   MIMIC-IV

The Medical Information Mart for Intensive Care-IV (MIMIC-IV) is a large collection of data for over 40,000 patients at intensive care units at the Beth Israel Deaconess Medical Center. It contains retrospectively collected medical data for 299,712 patients, 431,231 admissions, and 73,181 ICU stays. It improves upon the MIMIC-III dataset, incorporating more up-to-date medical data with an optimized data storage structure. In our study, we use vital signs for time-series generation. To simplify the data-cleaning process, we adopt scripts from the MIMIC Code Repository.

**Variable Selection:** We extracted five vital signs for each patient from MIMIC-IV. The selected variables are heart rate (beats per minute), systolic blood pressure (mm Hg), diastolic blood pressure (mm Hg), respiratory rate (breaths per minute), and oxygen saturation (%). We extract all measurements of each feature within the first 72 hours of each patient's ICU admission. Similar to MIMIC-III, we encode the features using the method described in Section 4.2 for model training.

**Cohort Selection:** Similar to MIMIC-III, we select our MIMIC-IV study cohort by applying filtering criteria provided by the official MIMIC-IV repository. The criteria can be accessed here. We also select patients at the unit stay level and include those who stayed for at least 72 hours in ICU.

We use 75% for training and 25% for testing, and the class ratio is kept similar across the training and testing data.

### A.1.4   EICU

The eICU Collaborative Research Database is a multi-center database with 200,859 admissions to intensive care units monitored by the eICU programs across the United States. It includes various information for the patients, such as vital sign measurements, care plan documentation, severity of illness measures, diagnosis information, and treatment information. The database contains 139,367 patients admitted to critical care units between 2014 and 2015.

**Variable Selection:** We select four vital sign variables from the *vitalPeriodic* table in our study: heart rate (beats per minute), respiratory rate (breaths per minute), oxygen saturation (%), and mean blood pressure (mm Hg). The measurements are recorded as one-minute averages and are then stored as five-minute medians. We extract values between each patient's first hour of the ICU stay and the next 24 hours for the selected variables. Since the measurements are recorded at 5-minute intervals, we obtain a multivariate time series of length 276 for each patient in our study cohort.

**Cohort Selection:** We select patients for our eICU study cohort by filtering the time interval. Specifically, we include patients who stay for at least 24 hours in their ICU stay, and the time series measurements are extracted. We did not use filtering criteria for time series in eICU. This is a design choice that allows us to evaluate TIMEDIFF when unfiltered time series are used as the input. We also select patients at the unit stay level.

We use 75% for training and 25% for testing while ensuring the class ratio is similar between the two data splits.

### A.1.5   HIRID

The high time resolution ICU dataset (HiRID) is a publicly accessible critical care dataset consisting of data for more than 33,000 admissions to the Department of Intensive Care Medicine of the Bern University Hospital, Switzerland. It includes de-identified demographic information and 712 physiological variables, diagnostic tests, and treatment information between January 2008 to June 2016. The physiological measurements are recorded at 2-minute intervals.

**Variable Selection:** We consider seven variables in our study: heart rate (beats per minute), invasive systolic arterial pressure (mm Hg), invasive diastolic arterial pressure (mm Hg), invasive mean arterial pressure (mm Hg), peripheral oxygen saturation (%), ST elevation (mm), and central venous

pressure (mm Hg). We selected the recorded data during the first 200 minutes of each patient's ICU stay.

**Cohort Selection**: We include patients who stayed for at least 200 minutes in our HiRID study cohort. Unlike all aforementioned EHR datasets, our HiRID study cohort only includes patients without missing values. This design choice allows us to evaluate the performance of TIMEDIFF in the absence of missing values on EHR datasets.

We use 80% of our study cohort as the training data and 20% as the testing data, and the mortality rate is kept similar between the splits.

## A.2 BASELINES

We reference the following source code for implementations of our baselines.

Table 6: Source code links for all baselines.

| Method | Source Code Link |
|---|---|
| EHR-M-GAN (Li et al., 2023) | LINK |
| DSPD/CSPD (GP or OU) (Biloš et al., 2023) | LINK |
| GT-GAN (Jeon et al., 2022) | LINK |
| TimeGAN (Yoon et al., 2019) | LINK |
| RCGAN (Esteban et al., 2017) | LINK |
| C-RNN-GAN (Mogren, 2016) | LINK |
| T-Forcing (Graves, 2013; Sutskever et al., 2011) | LINK |
| P-Forcing (Lamb et al., 2016) | LINK |

## A.3 MODEL TRAINING AND HYPERPARAMETER SELECTION

### A.3.1 NEURAL CONTROLLED DIFFERENTIAL EQUATION

We attempted to use neural controlled differential equation (NCDE) (Kidger et al., 2020) as our architecture for $s_\theta$. We expect the continuous property of the NCDE to yield better results for time-series generation. NCDE is formally defined as the following:

**Definition 1** *Suppose we have a time-series $s = \{(r_1, \boldsymbol{x}_1), ..., (r_n, \boldsymbol{x}_n)\}$ and $D$ is the dimensionality of the series. Let $Y : [r_1, r_n] \to \mathbb{R}^{D+1}$ be the natural cubic spline with knots at $r_1, ..., r_n$ such that $Y_{t_i} = (\boldsymbol{x}_i, r_i)$. $s$ is often assumed to be a discretization of an underlying process that is approximated by $Y$. Let $f_\theta : \mathbb{R}^h \to \mathbb{R}^{h \times (D+1)}$ and $\zeta_\theta : \mathbb{R}^{D+1} \to \mathbb{R}^h$ be any neural networks, where $h$ is the size of hidden states. Let $z_{r_1} = \zeta_\theta(r_1, \boldsymbol{x}_1)$*

*The NCDE model is then defined to be the solution to the following CDE:*

$$z_r = z_{r_1} + \int_{r_1}^{r} f_\theta(z_s) \mathrm{d}Y_s \quad \text{for } r \in (r_1, r_n] \tag{14}$$

*where the integral is a Riemann–Stieltjes integral.*

However, we find that this approach suffers from high computational cost since it needs to calculate cubic Hermite spline and solve the CDE for every noisy sample input during training. It thus has low scalability for generating time-series data with long sequences. Nevertheless, we believe this direction is worth exploring for future research.

### A.3.2 TIMEDIFF TRAINING

The diffusion model is trained using $\mathcal{L}_{\text{train}}$ in Equation (12). We set $\lambda$ to 0.01. We use cosine scheduling (Nichol & Dhariwal, 2021b) for the variances $\left\{\beta^{(t)}\right\}_{t=1}^{T}$. We apply the exponential moving average to model parameters with a decay rate of 0.995. We use Adam optimizer (Kingma & Ba, 2015) with a learning rate of 0.00008, $\beta_1 = 0.9$, and $\beta_2 = 0.99$. We set the total diffusion step $T$ to be 1000, accumulate the gradient for every 2 steps, use 2 layers for the BRNN, and use a batch size of 32 across all our experiments.

### A.3.3 BASELINES

For a fair comparison, we use a 2-layer RNN with a hidden dimension size of four times the number of input features. We utilize the LSTM as our architecture whenever applicable. We use a hidden dimension size of 256 for the eICU dataset.

For deterministic models such as the T-Forcing and P-Forcing, we uniformly sample the initial data vector from the real training data. We subsequently use the initial data vector as an input to the deterministic models to generate the synthetic sequence by unrolling.

For stochastic process diffusion, we set *gp_sigma* to be 0.1 for Gaussian process (GP) and *ou_theta* to be 0.5 for Ornstein-Uhlenbeck (OU) process. For discrete diffusion, we set the total diffusion step at 1000. We use Adam optimizer with a learning rate of 0.00001 and batch size of 32 across all the experiments.

### A.3.4 SOFTWARE

We set the seed to 2023 and used the following software for our experiments.

Table 7: Software packages.

| Method | Software |
|---|---|
| TIMEDIFF | PyTorch 2.0.1 |
| EHR-M-GAN (Li et al., 2023) | TensorFlow 1.14.0 |
| DSPD/CSPD (GP or OU) (Biloš et al., 2023) | PyTorch 2.0.1 |
| GT-GAN (Jeon et al., 2022) | PyTorch 2.0.0 |
| TimeGAN (Yoon et al., 2019) | TensorFlow 1.10.0 |
| RCGAN (Esteban et al., 2017) | TensorFlow 1.10.0 |
| C-RNN-GAN (Mogren, 2016) | PyTorch 2.0.1 |
| T-Forcing (Graves, 2013; Sutskever et al., 2011) | PyTorch 1.0.0 |
| P-Forcing (Lamb et al., 2016) | PyTorch 1.0.0 |

### A.4 EVALUATION METRICS

### A.4.1 DISCRIMINATIVE AND PREDICTIVE SCORES

To ensure consistency with results obtained from TimeGAN and GT-GAN, we adopt the same source code from TimeGAN for calculating discriminative scores. We train a GRU time-series classification model to distinguish between real and synthetic samples, and $|0.5 - \text{Accuracy}|$ is used as the score.

For predictive scores, we use the implementation from GT-GAN, which computes the mean absolute error based on the next step *vector* prediction (see Appendix D of the GT-GAN paper (Jeon et al., 2022)). For consistency, we compute the predictive scores for the Stocks and Energy datasets by employing the implementation from TimeGAN that calculates the error for the next step *scalar* prediction. We apply standardization to the inputs of the discriminator and predictor and use linear activation for the predictor for all EHR datasets.

### A.4.2 T-SNE

We perform hyperparameter search on the number of iterations, learning rate, and perplexity to optimize the performance of t-SNE (Wattenberg et al., 2016). We use 300 iterations, perplexity 30, and scaled learning rate (Belkina et al., 2019). We flatten the input data along the feature dimension, perform standardization, and then apply t-SNE directly to the data without using any summary statistics. We uniformly randomly select 2000 samples from the synthetic, real training, and real testing data for t-SNE visualizations on the eICU, MIMIC-III, MIMIC-IV, and Energy datasets. For the HiRID and Stocks dataset, we use 1000 and 700 samples, respectively, due to the limited size of real testing data.

### A.4.3 IN-HOSPITAL MORTALITY PREDICTION

**Train on Synthetic, Test on Real (TSTR):** We use the default hyperparameters for the six ML models described in Section 5 using the scikit-learn software package. The models are trained using two input formats: (1) raw multivariate time-series data flattened along the feature dimension; (2) summary statistics for each feature (the first record since ICU admission, minimum, maximum, record range, mean, standard deviation, mode, and skewness). After training, the models are evaluated on real testing data in terms of AUC.

**Train on Synthetic and Real, Test on Real (TSRTR):** To evaluate the effect of the increased proportion of the synthetic samples for training on model performance, we uniformly randomly sample 2,000 real training data from our training set and use this subset to train TIMEDIFF. After training of TIMEDIFF is complete, we subsequently add different amounts of the synthetic samples to the 2,000 real samples to train ML models for in-hospital mortality prediction. We set the synthetic percentages to be 0.1, 0.3, 0.5, 0.7, 0.9. In other words, the ML models are trained with at most 20,000 samples (18,000 synthetic and 2,000 real). This evaluation also simulates the scenario where synthetic samples from TIMEDIFF are used for data augmentation tasks in medical applications. Similar to computing the TSTR score, we train the ML models using either raw time-series data or summary statistics of each feature as the input. Results obtained using summary statistics as the input are presented in Appendix B.4.

### A.4.4 NNAA RISK

We calculate the NNAA risk score (Yale et al., 2020) by using the implementation from this repository. Similar to performing t-SNE, we flatten the data along the feature dimension and apply standardization for preprocessing. The scaled datasets are then used to calculate the NNAA risk score.

For reference, we describe the components of the NNAA score below.

**Definition 2** *Let $S_T = \{x_T^{(1)}, ..., x_T^{(n)}\}$, $S_E = \{x_E^{(1)}, ..., x_E^{(n)}\}$ and $S_S = \{x_S^{(1)}, ..., x_S^{(n)}\}$ be data samples with size $n$ from real training, real testing, and synthetic datasets, respectively. The NNAA risk is the difference between two accuracies:*

$$NNAA = AA_{\text{test}} - AA_{\text{train}}, \tag{15}$$

$$AA_{\text{test}} = \frac{1}{2}\left(\frac{1}{n}\sum_{i=1}^{n}\mathbb{I}\{d_{ES}(i) > d_{EE}(i)\} + \frac{1}{n}\sum_{i=1}^{n}\mathbb{I}\{d_{SE}(i) > d_{SS}(i)\}\right), \tag{16}$$

$$AA_{\text{train}} = \frac{1}{2}\left(\frac{1}{n}\sum_{i=1}^{n}\mathbb{I}\{d_{TS}(i) > d_{TT}(i)\} + \frac{1}{n}\sum_{i=1}^{n}\mathbb{I}\{d_{ST}(i) > d_{SS}(i)\}\right), \tag{17}$$

*where $\mathbb{I}\{\cdot\}$ is the indicator function, and*

$$d_{TS}(i) = \min_j \left\|x_T^{(i)} - x_S^{(j)}\right\|, \quad d_{ST}(i) = \min_j \left\|x_S^{(i)} - x_T^{(j)}\right\|, \tag{18}$$

$$d_{ES}(i) = \min_j \left\|x_E^{(i)} - x_S^{(j)}\right\|, \quad d_{SE}(i) = \min_j \left\|x_S^{(i)} - x_E^{(j)}\right\|, \tag{19}$$

$$d_{TT}(i) = \min_{j, j \neq i} \left\|x_T^{(i)} - x_T^{(j)}\right\|, \quad d_{SS}(i) = \min_{j, j \neq i} \left\|x_S^{(i)} - x_S^{(j)}\right\|, \quad d_{EE}(i) = \min_{j, j \neq i} \left\|x_E^{(i)} - x_E^{(j)}\right\|. \tag{20}$$

In our experiments, there are instances where $AA_{\text{train}} > AA_{\text{test}}$. To consistently obtain positive values, we use NNAA $= |AA_{\text{test}} - AA_{\text{train}}|$ for our evaluations.

### A.4.5 MIR

Our implementation of the MIR score (Liu et al., 2019) follows the source code in this repository. To keep a similar scale of the distance across different datasets, we apply normalization on the computed distances so that they are in the [0,1] range. We use a threshold of 0.08 for the MIMIC-IV, MIMIC-III, and HiRID datasets. We set the decision threshold to 0.005 for eICU. All the input data is normalized to the [0,1] range.

# B ADDITIONAL EXPERIMENTS

## B.1 T-SNE VISUALIZATIONS

We present our visualizations for all the baselines in our experiments in this section. For all the figures, synthetic samples are in **blue**, real samples in train split are in **red**, and real samples in test split are in **orange**. We discuss our procedure for t-SNE visualizations in Appendix A.4.2.

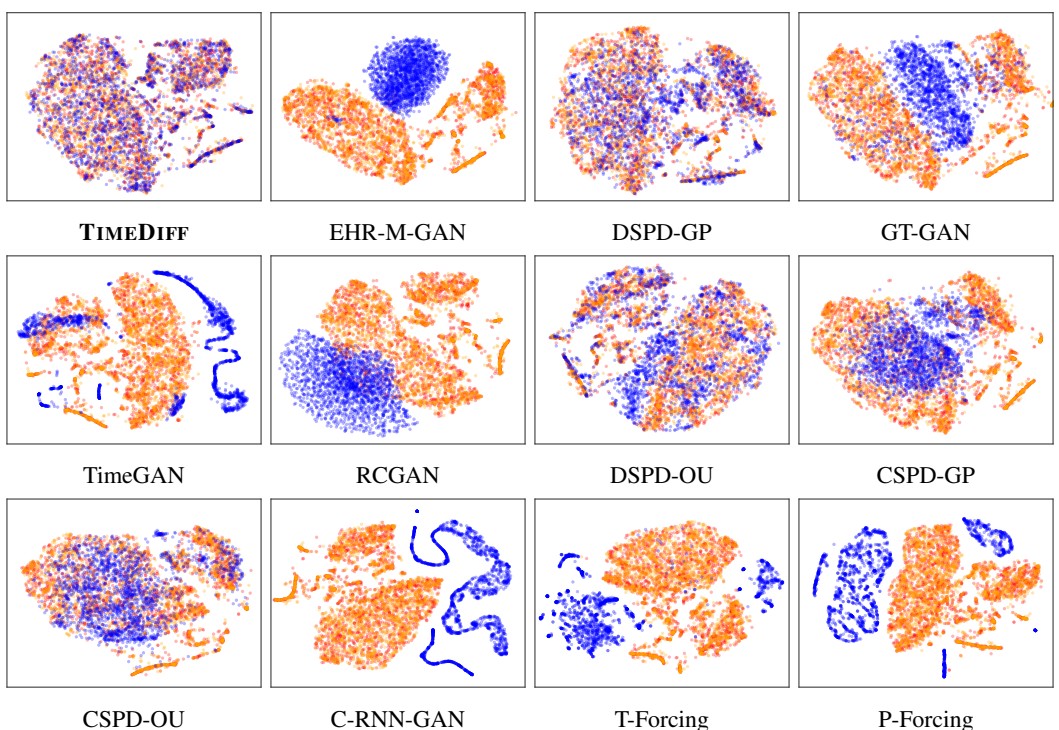

|  |  |  |  |
|---|---|---|---|
| **TIMEDIFF** | EHR-M-GAN | DSPD-GP | GT-GAN |
| TimeGAN | RCGAN | DSPD-OU | CSPD-GP |
| CSPD-OU | C-RNN-GAN | T-Forcing | P-Forcing |

Figure 3: t-SNE dimension reduction visualization for eICU.

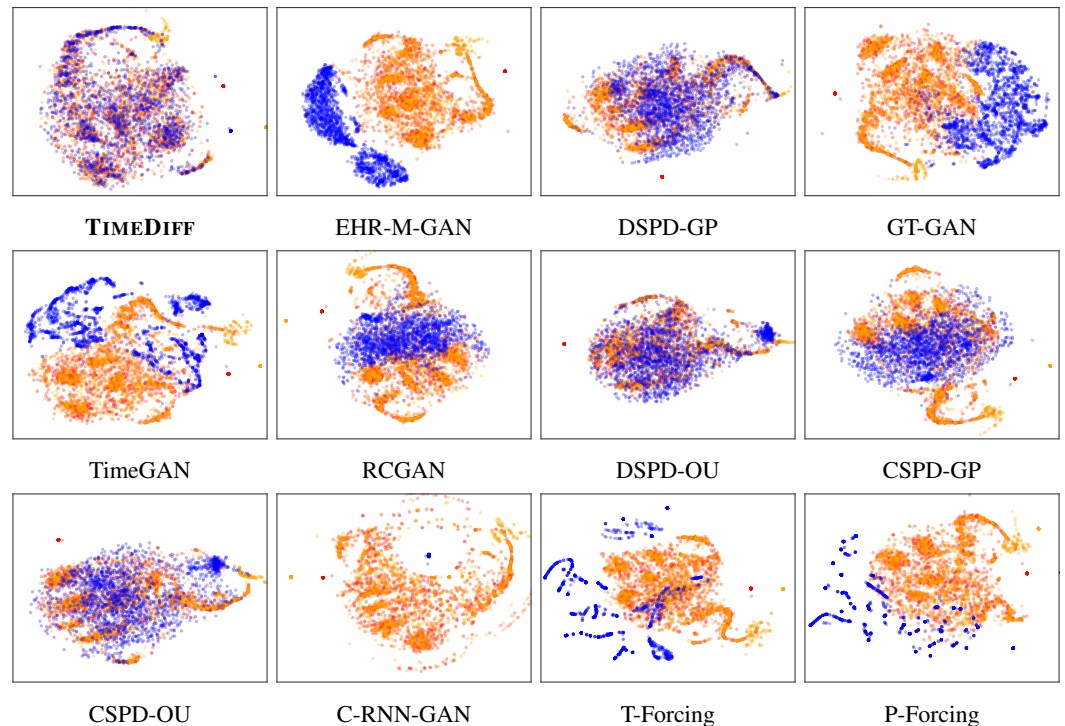

Figure 4: t-SNE dimension reduction visualization for MIMIC-III.

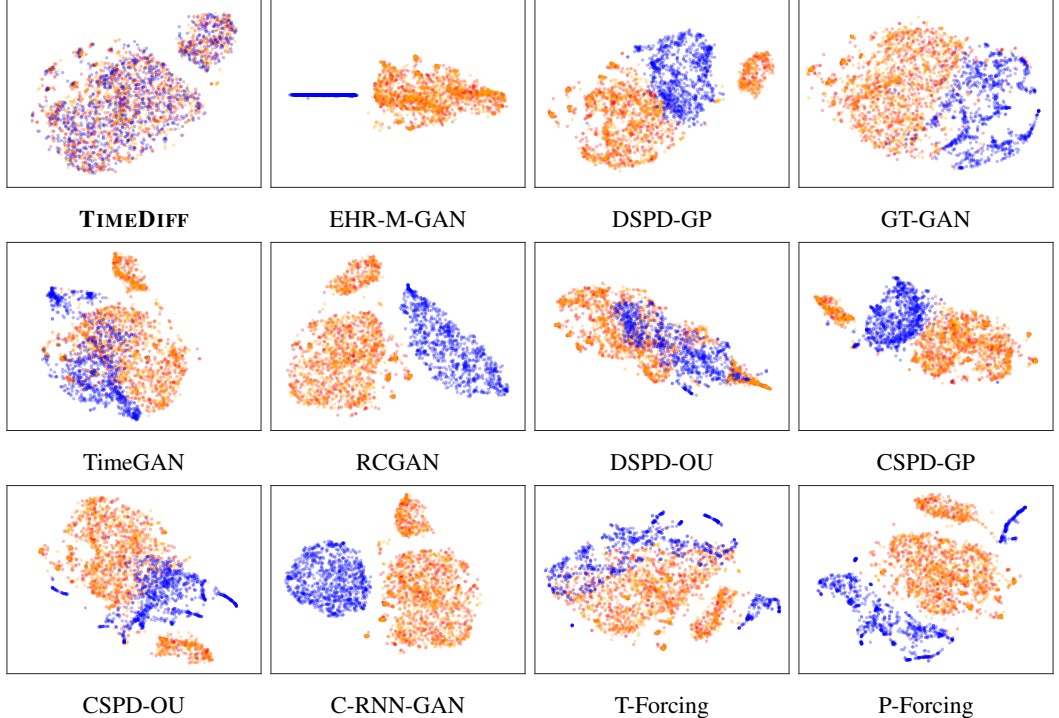

Figure 5: t-SNE dimension reduction visualization for HiRID.

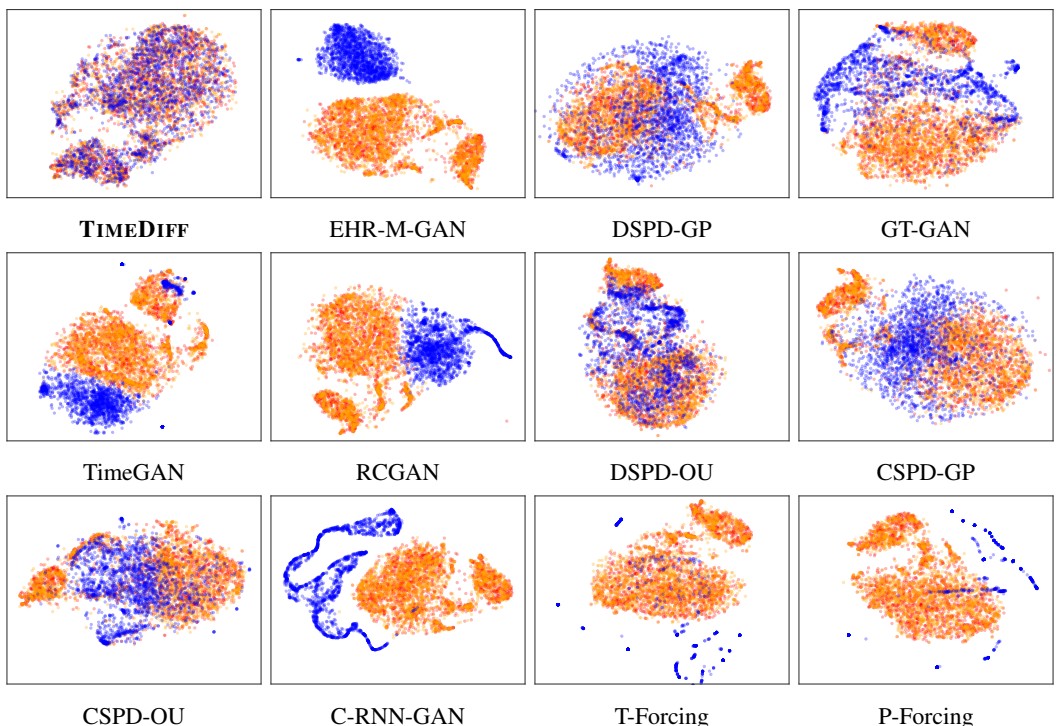

Figure 6: t-SNE dimension reduction visualization for MIMIC-IV.

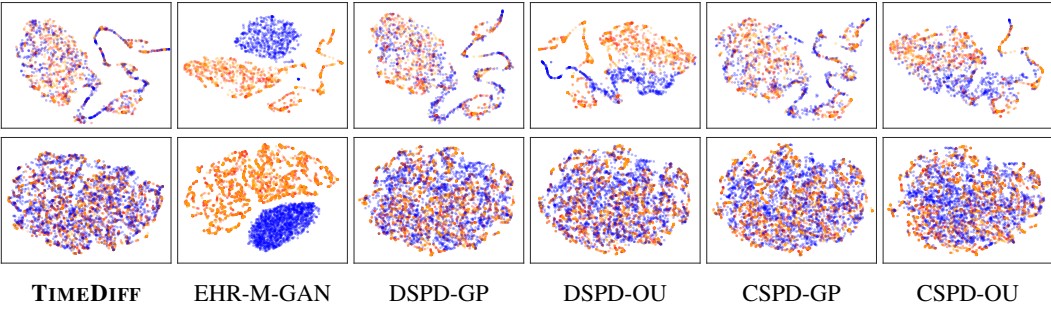

Figure 7: t-SNE dimension reduction visualization for non-EHR datasets. First row is Stocks and second row is Energy.

### B.2 RUNTIME COMPARISONS

In this section, we present additional runtime comparisons across all EHR datasets. We consider EHR-M-GAN, stochastic process diffusion models, TimeGAN, and GT-GAN. We use Intel Xeon Gold 6226 Processor and Nvidia GeForce 2080 RTX Ti to train all the models for a fair comparison.

Table 8: Comparison of runtime (hours).

| Dataset | **TIMEDIFF** | EHR-M-GAN | TimeGAN | GT-GAN | DSPD-GP | DSPD-OU | CSPD-GP | CSPD-OU |
|---|---|---|---|---|---|---|---|---|
| MIMIC-III | 2.7 | 18.9 | 10.8 | 21.8 | 2.5 | 2.5 | 2.5 | 2.5 |
| MIMIC-IV | 2.7 | 28.8 | 29.5 | 47.3 | 2.6 | 2.6 | 2.6 | 2.6 |
| HiRID | 2.5 | 29.7 | 46.2 | 58.3 | 2.8 | 2.8 | 2.8 | 2.8 |
| eICU | 8.7 | 87.1 | 110 | 59.1 | 7.0 | 7.0 | 7.0 | 7.0 |

### B.3 EFFECT OF $\lambda$

In this section, we investigate the effect of hyperparameter $\lambda$ on TIMEDIFF. We trained TIMED-IFF using $\lambda \in \{0.001, 0.01, 0.1, 1, 10\}$ while keeping the other hyperparameters identical as those described in Appendix A.3.2.

Table 9: Effect of $\lambda$ on data utility.

| Metric | Method | MIMIC-III | MIMIC-IV | eICU |
|---|---|---|---|---|
| Discriminative Score ($\downarrow$) | $\lambda = 0.001$ | .106±.047 | .054±.023 | .018±.010 |
| | $\lambda = 0.01$ | .028±.023 | .030±.022 | .015±.007 |
| | $\lambda = 0.1$ | .045±.046 | .036±.026 | .027±.011 |
| | $\lambda = 1$ | .108±.041 | .125±.068 | .068±.016 |
| | $\lambda = 10$ | .430±.037 | .441±.090 | .299±.048 |
| | *Real Data* | *.012±.006* | *.014±.011* | *.004±.003* |
| Predictive Score ($\downarrow$) | $\lambda = 0.001$ | .472±.002 | .433±.002 | .305±.017 |
| | $\lambda = 0.01$ | .469±.003 | .432±.002 | 309±.019 |
| | $\lambda = 0.1$ | .469±.002 | .434±.002 | .319±.036 |
| | $\lambda = 1$ | .472±.003 | .435±.002 | .317±.036 |
| | $\lambda = 10$ | .496±.002 | .488±.008 | .314±.018 |
| | *Real Data* | *.467±.005* | *.433±.001* | *.304±.017* |

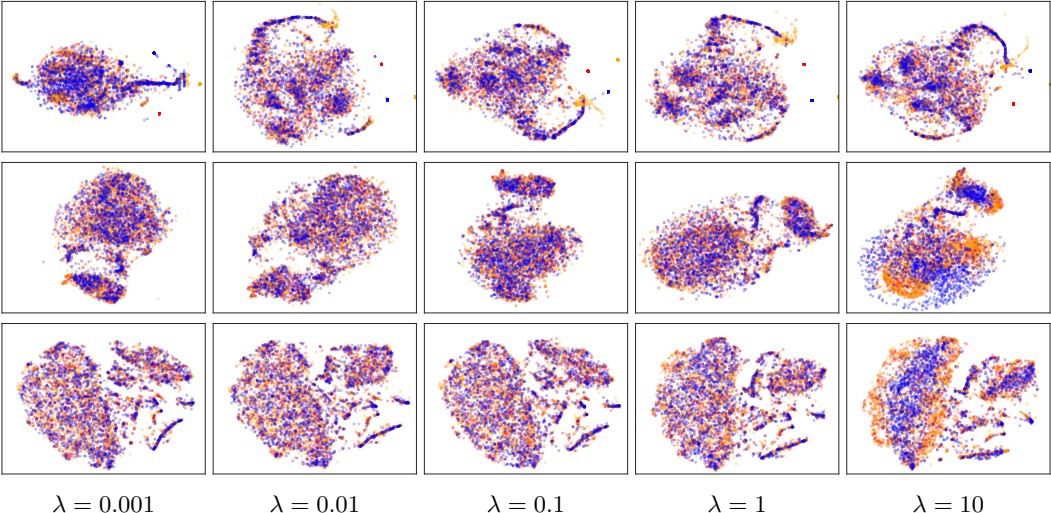

|  $\lambda = 0.001$ | $\lambda = 0.01$ | $\lambda = 0.1$ | $\lambda = 1$ | $\lambda = 10$ |

Figure 8: t-SNE visualizations. First row is MIMIC-III, second row is MIMIC-IV, and third row is eICU dataset.

### B.4 TSTR/TSRTR AND PRIVACY RISK EVALUATIONS

In this section, we provide additional results for TSTR and TSRTR scores across all four EHR datasets we considered in this study. We train ML models using one of two methods: flattening the feature dimension of raw time series data, or using summary statistics such as initial measurement, minimum, maximum, range, mean, standard deviation, mode, and skewness.

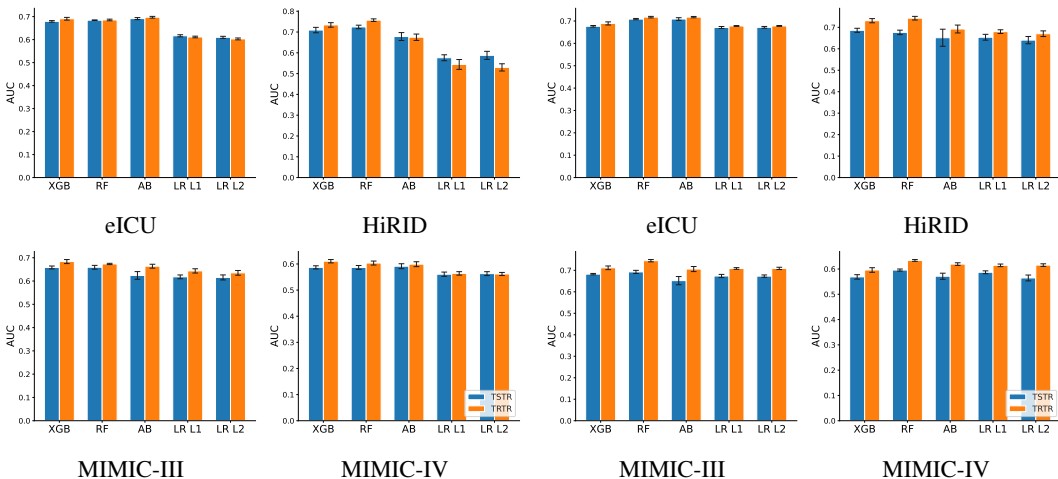

(a) ML models are trained with raw time series values (with flattened feature dimension).

(b) ML models are trained with summary statistics.

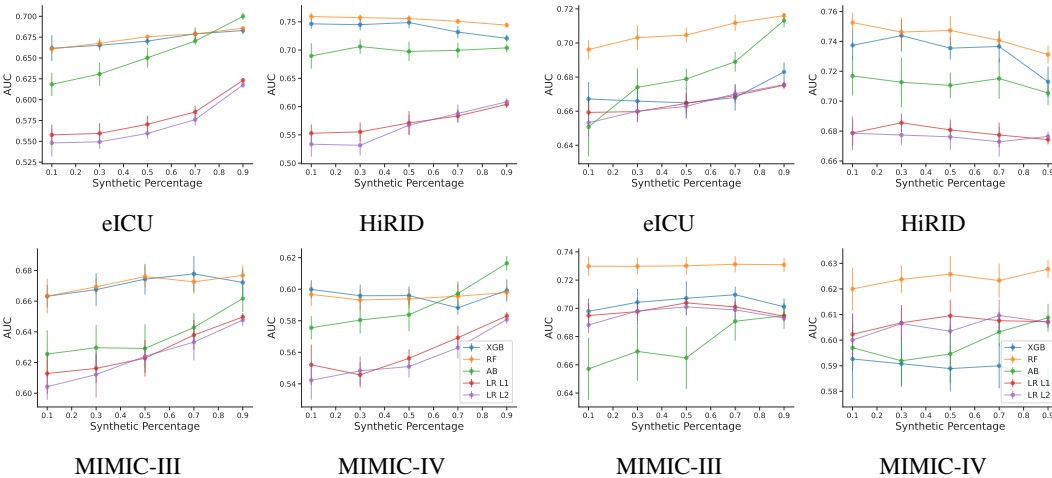

(a) ML models are trained with raw time series values (with flattened feature dimension).

(b) ML models are trained with summary statistics.

As an additional evaluation, we train RNN classifiers on synthetic time series data from TIMEDIFF and evaluate their performance on real testing data. We use bidirectional RNNs with a hidden dimension of 64 for this experiment.

Table 10: TSTR and TRTR scores for RNN classifiers.

| Method | Metric | MIMIC-III | MIMIC-IV | HiRID | eICU |
|--------|--------|-----------|----------|-------|------|
| GRU | TSTR | .584±.016 | .516±.025 | .509±.042 | .544±.020 |
| | TRTR | .543±.018 | .507±.022 | .463±.050 | .476±.029 |
| LSTM | TSTR | .581±.019 | .484±.010 | .502±.061 | .558±.037 |
| | TRTR | .587±.026 | .473±.025 | .420±.051 | .531±.029 |

Table 11: Full results for privacy risk evaluations.

| Metric | Method | MIMIC-III | MIMIC-IV | HiRID | eICU |
|---|---|---|---|---|---|
| $AA_{\text{test}}$ (∼0.5) | **TIMEDIFF** | **.574±.002** | **.517±.002** | **.531±.003** | **.537±.001** |
| | EHR-M-GAN | .998±.000 | 1.000±.000 | 1.000±.000 | .977±.000 |
| | DSPD-GP | .974±.001 | .621±.002 | .838±.004 | .888±.000 |
| | DSPD-OU | .927±.000 | .804±.003 | .886±.001 | .971±.000 |
| | CSPD-GP | .944±.001 | .623±.002 | .958±.002 | .851±.001 |
| | CSPD-OU | .967±.001 | .875±.002 | .947±.001 | .982±.000 |
| | GT-GAN | .995±.000 | .910±.001 | .990±.001 | .981±.000 |
| | TimeGAN | .997±.000 | .974±.001 | .643±.003 | 1.000±.000 |
| | RCGAN | .983±.001 | .999±.000 | 1.000±.000 | 1.000±.000 |
| | C-RNN-GAN | 1.000±.000 | .993±.000 | 1.000±.000 | 1.000±.000 |
| | T-Forcing | 1.000±.000 | .928±.001 | .946±.001 | .999±.000 |
| | P-Forcing | 1.000±.000 | .977±.001 | .998±.000 | 1.000±.000 |
| | *Real Data* | *.552±.002* | *.497±.002* | *.511±.006* | *.501±.002* |
| $AA_{\text{train}}$ (∼0.5) | **TIMEDIFF** | **.573±.002** | **.515±.002** | **.531±.002** | **.531±.002** |
| | EHR-M-GAN | .999±.000 | 1.000±.000 | 1.000±.000 | .965±.002 |
| | DSPD-GP | .968±.002 | .620±.003 | .851±.005 | .888±.001 |
| | DSPD-OU | .928±.001 | .788±.003 | .876±.002 | .971±.000 |
| | CSPD-GP | .940±.002 | .629±.005 | .965±.004 | .852±.001 |
| | CSPD-OU | .966±.001 | .880±.003 | .945±.002 | .983±.000 |
| | GT-GAN | .995±.001 | .907±.002 | .989±.001 | .981±.000 |
| | TimeGAN | .997±.000 | .969±.003 | .651±.004 | 1.000±.000 |
| | RCGAN | .984±.001 | .999±.000 | 1.000±.000 | 1.000±.000 |
| | C-RNN-GAN | 1.000±.000 | .992±.001 | 1.000±.000 | 1.000±.000 |
| | T-Forcing | 1.000±.000 | .927±.002 | .941±.001 | .999±.000 |
| | P-Forcing | 1.000±.000 | .976±.002 | .998±.000 | 1.000±.000 |
| | *Real Data* | *.286±.003* | *.268±.004* | *.327±.006* | *.266±.002* |
| NNAA (↓) | **TIMEDIFF** | .002±.002 | .002±.002 | .004±.003 | .006±.002 |
| | EHR-M-GAN | .000±.000 | .000±.000 | .000±.000 | .012±.003 |
| | DSPD-GP | .005±.003 | .003±.003 | .013±.007 | .001±.001 |
| | DSPD-OU | .001±.001 | .016±.004 | .010±.002 | .000±.000 |
| | CSPD-GP | .004±.002 | .007±.005 | .008±.004 | .001±.001 |
| | CSPD-OU | .001±.001 | .005±.003 | .002±.001 | .001±.001 |
| | GT-GAN | .001±.000 | .004±.002 | .001±.001 | .000±.000 |
| | TimeGAN | .000±.000 | .005±.003 | .008±.004 | .000±.000 |
| | RCGAN | .001±.000 | .000±.000 | .000±.000 | .000±.000 |
| | C-RNN-GAN | .000±.000 | .001±.000 | .000±.000 | .000±.000 |
| | T-Forcing | .000±.000 | .002±.001 | .005±.001 | .000±.000 |
| | P-Forcing | .000±.000 | .002±.002 | .000±.000 | .000±.000 |
| | *Real Data* | *.267±.004* | *.229±.003* | *.184±.006* | *.235±.003* |
| MIR (↓) | **TIMEDIFF** | .191±.008 | .232±.048 | .236±.179 | .227±.021 |
| | EHR-M-GAN | .025±.007 | .435±.031 | .459±.161 | .049±.006 |
| | DSPD-GP | .032±.021 | .050±.009 | .106 ±.064 | .000±.000 |
| | DSPD-OU | .060±.032 | .007±.006 | .005±.005 | .000±.000 |
| | CSPD-GP | .060±.028 | .034±.017 | .004±.004 | .000±.000 |
| | CSPD-OU | .066±.046 | .016±.020 | .005±.003 | .000±.000 |
| | GT-GAN | .005±.002 | .046±.013 | .109±.057 | .000±.000 |
| | TimeGAN | .010±.002 | .173±.020 | .624±.006 | .000±.000 |
| | RCGAN | .013±.002 | .277±.049 | .063±.013 | .000±.000 |
| | C-RNN-GAN | .015±.005 | .011±.006 | .019±.005 | .000±.000 |
| | T-Forcing | .007±.003 | .215±.052 | .292±.125 | .000±.000 |
| | P-Forcing | .004±.004 | .131±.045 | .362±.233 | .003±.001 |
| | *Real Data* | *.948±.000* | *.929±.005* | *.737±.011* | *.927±.001* |

## B.5 REAL AND SYNTHETIC TIME SERIES VISUALIZATIONS

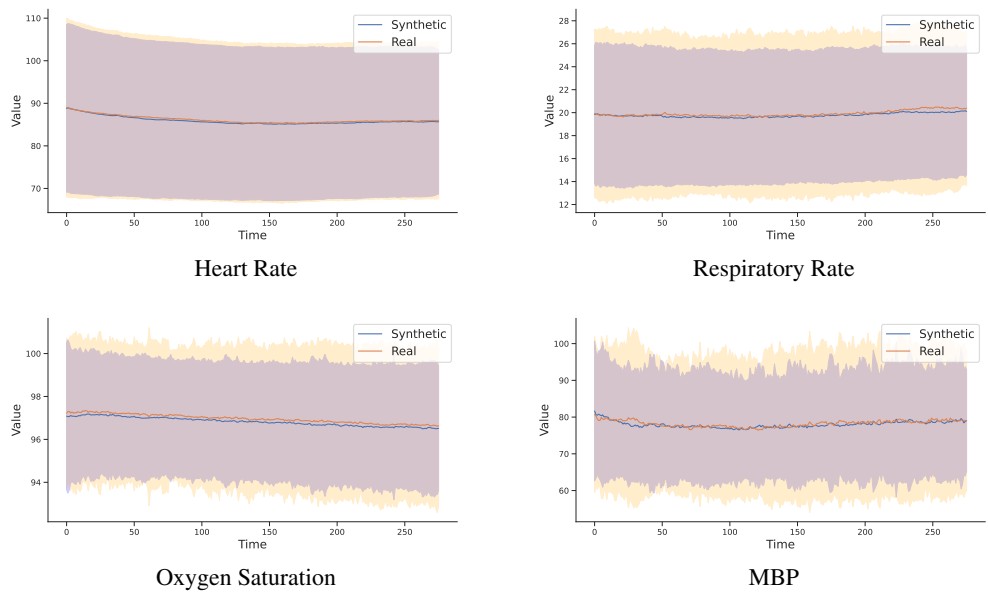

Figure 11: eICU, where the mean is the solid line and $\pm$ one standard deviation is the shaded area.

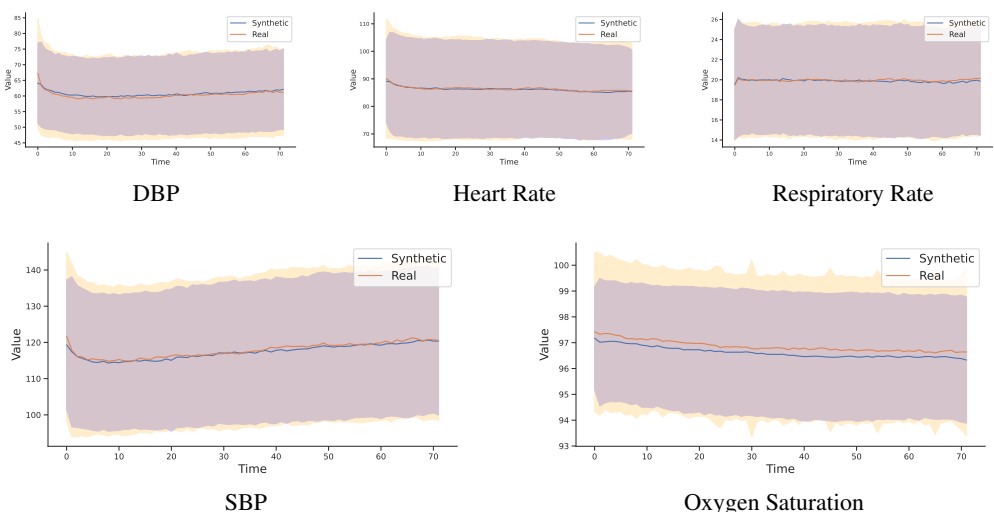

Figure 12: MIMIC-IV, where the mean is the solid line and $\pm$ one standard deviation is the shaded area.

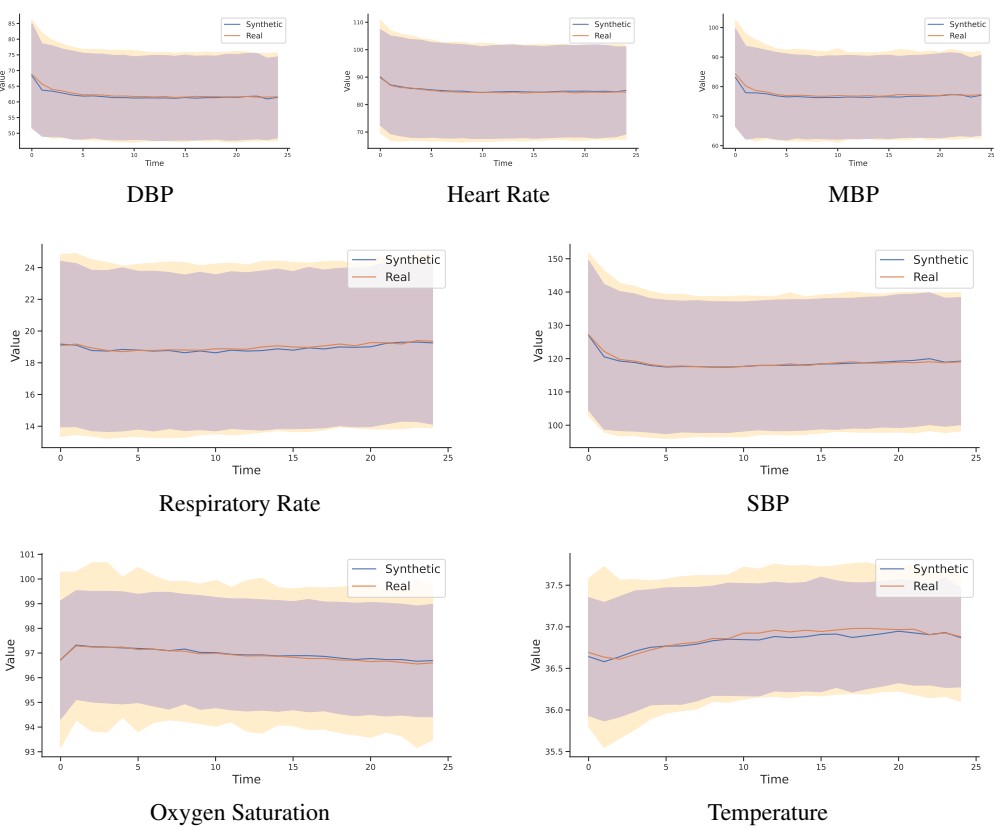

Figure 13: MIMIC-III, where the mean is the solid line and $\pm$ one standard deviation is the shaded area.

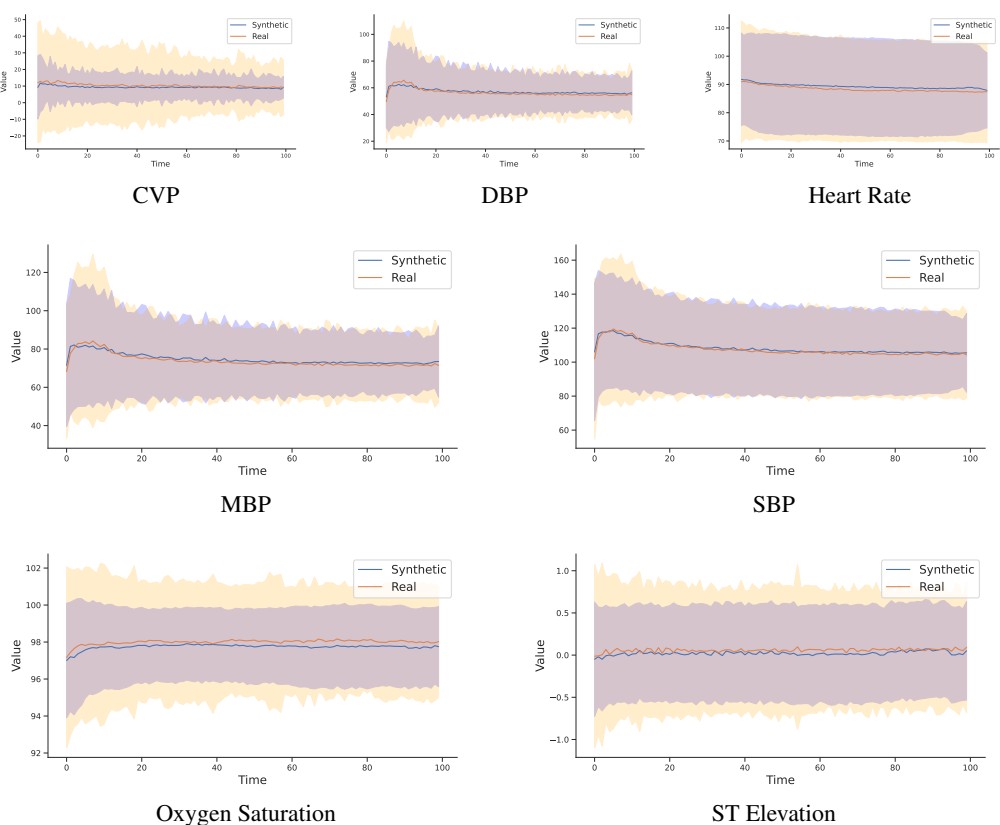

Figure 14: HiRID, where the mean is the solid line and $\pm$ one standard deviation is the shaded area.

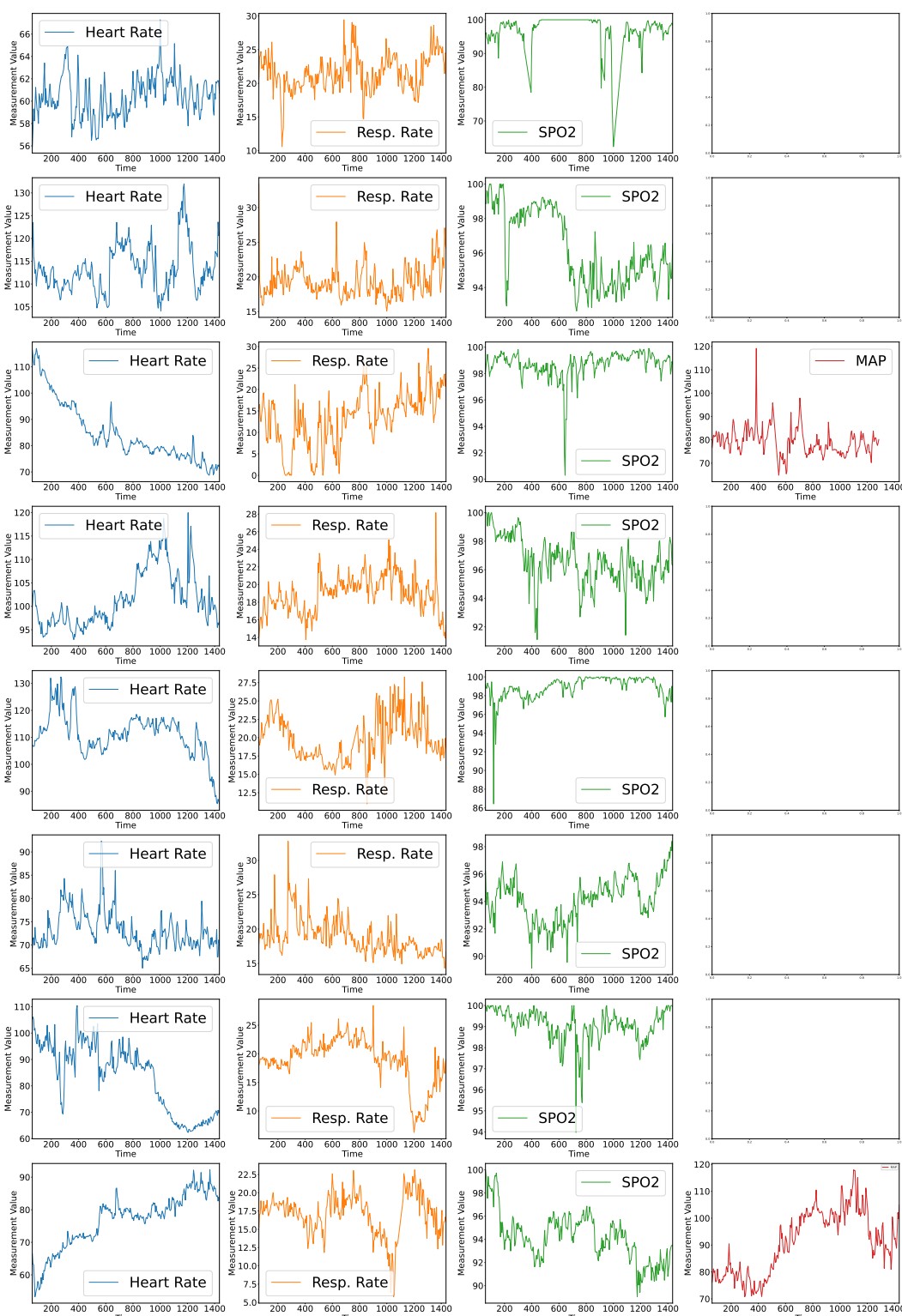

eICU: synthetic time series produced by TIMEDIFF.

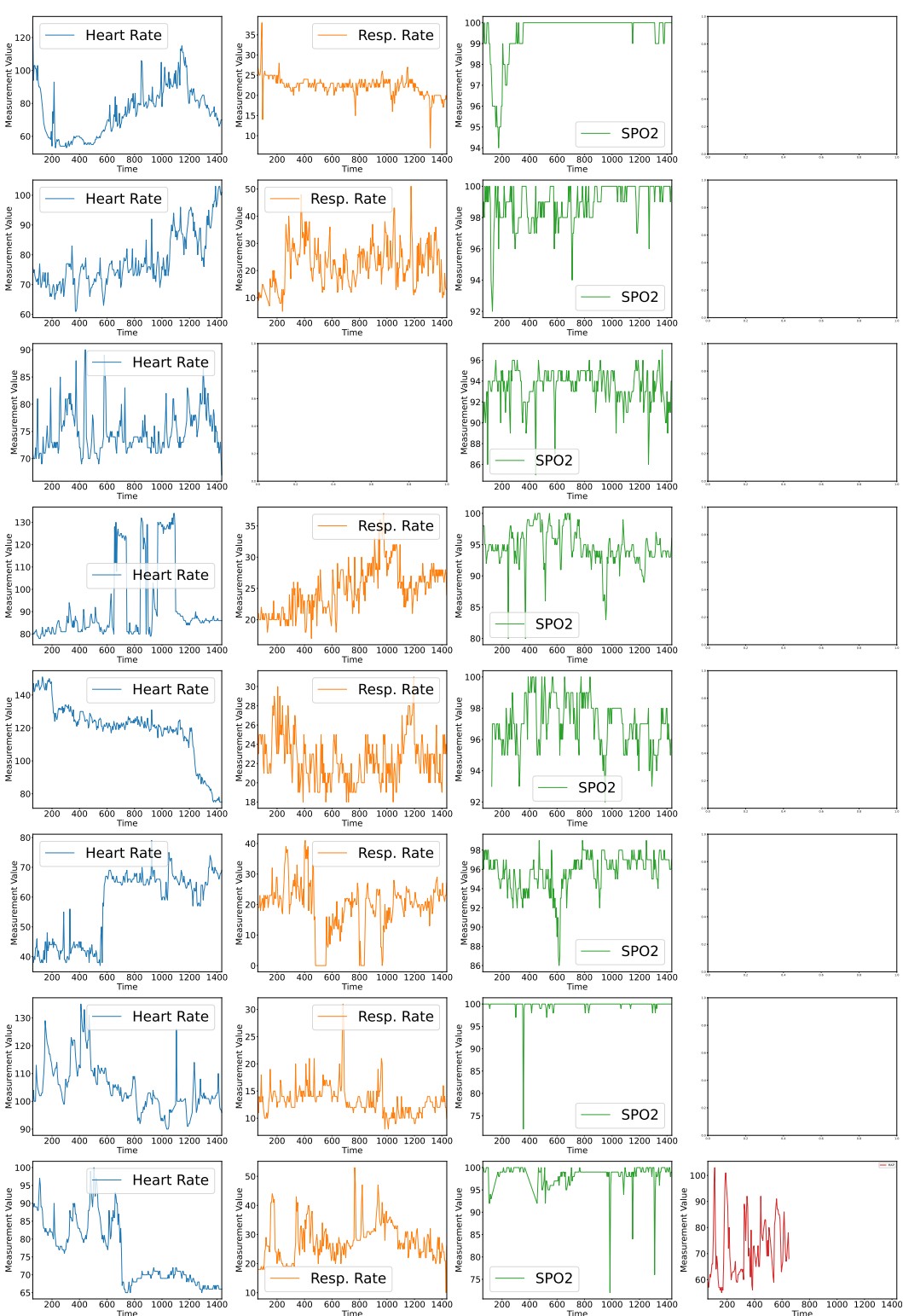

eICU: time series in real testing data.

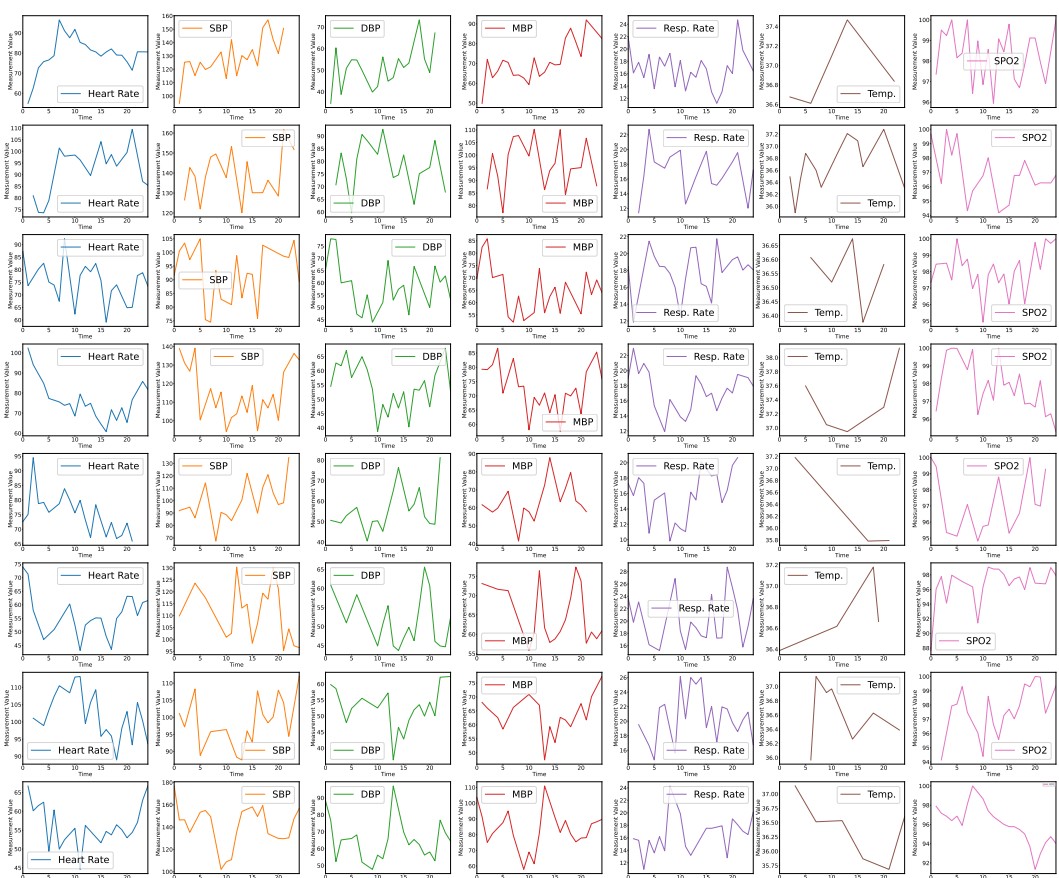

MIMIC-III: synthetic time series produced by TIMEDIFF.

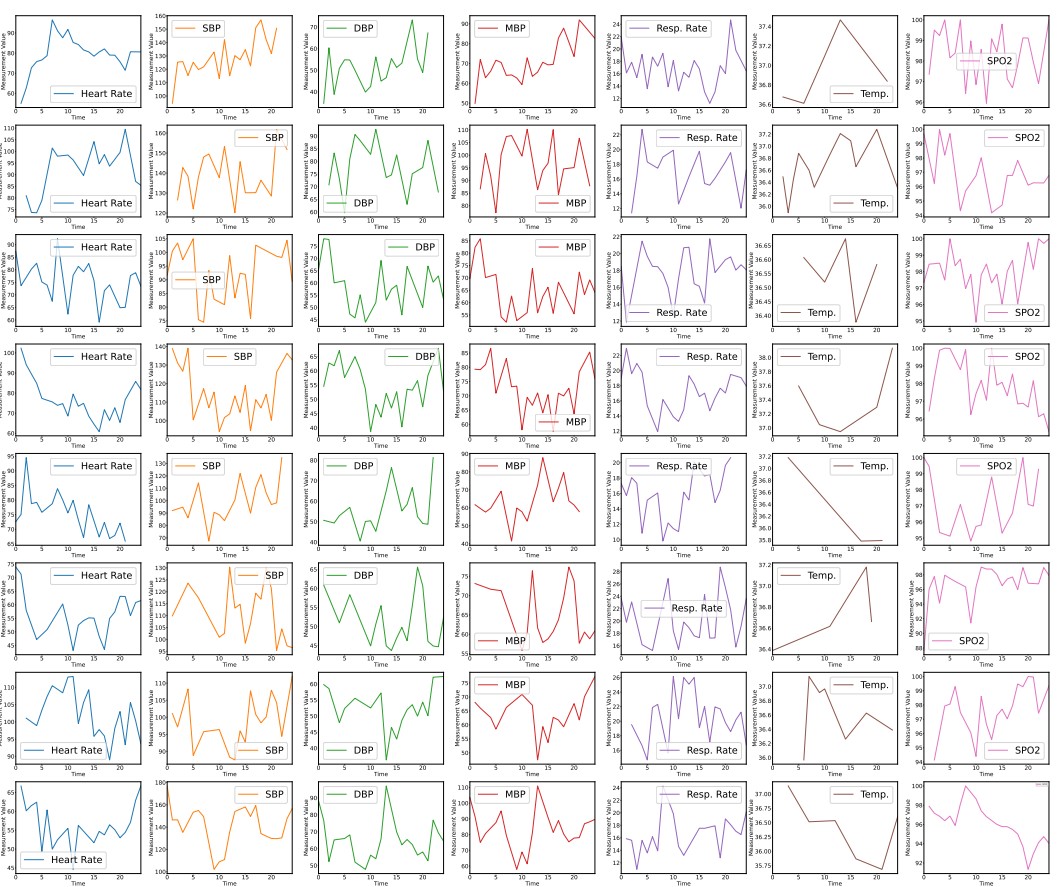

MIMIC-III: time series in real testing data.

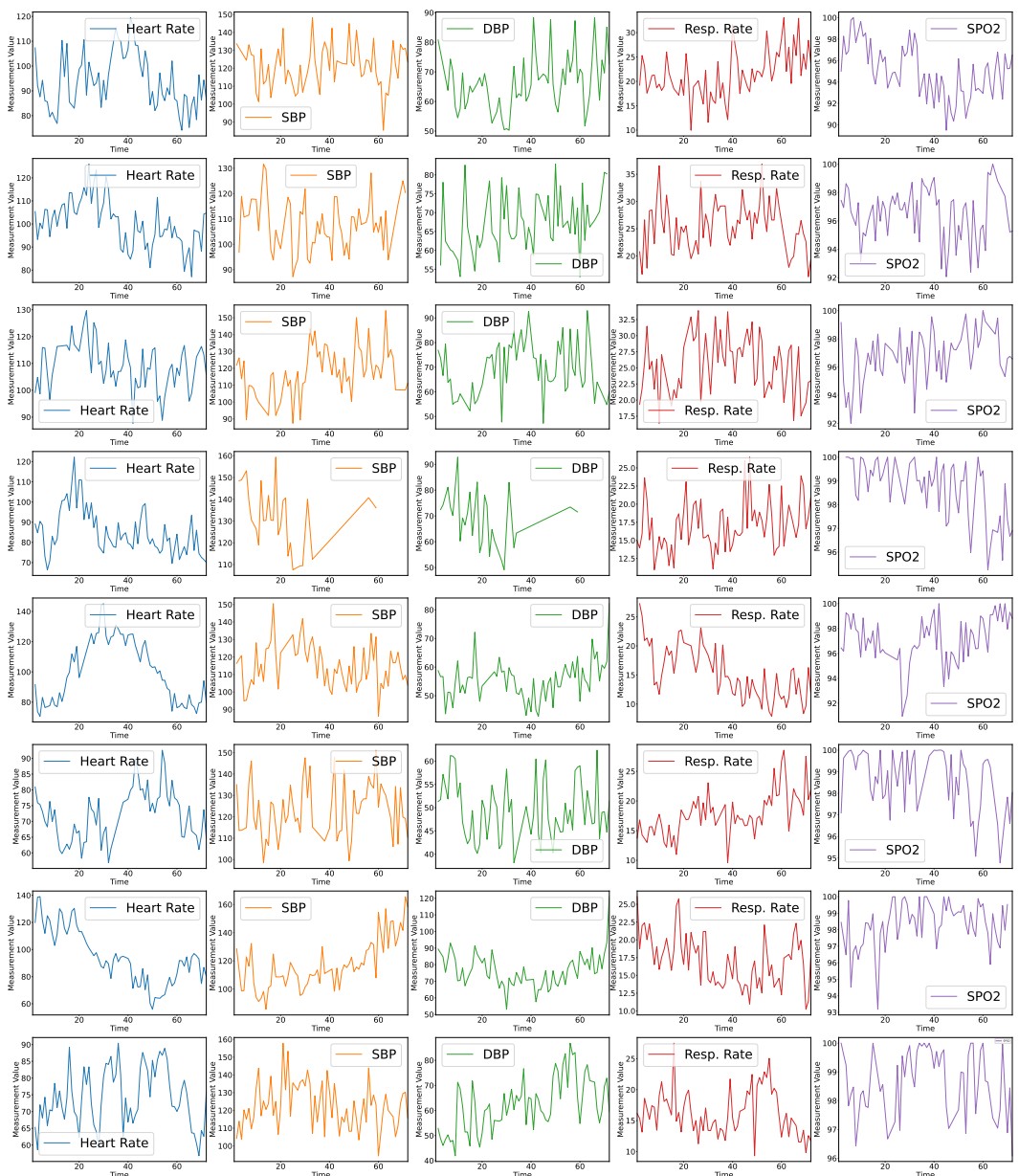

MIMIC-IV: synthetic time series produced by TIMEDIFF.

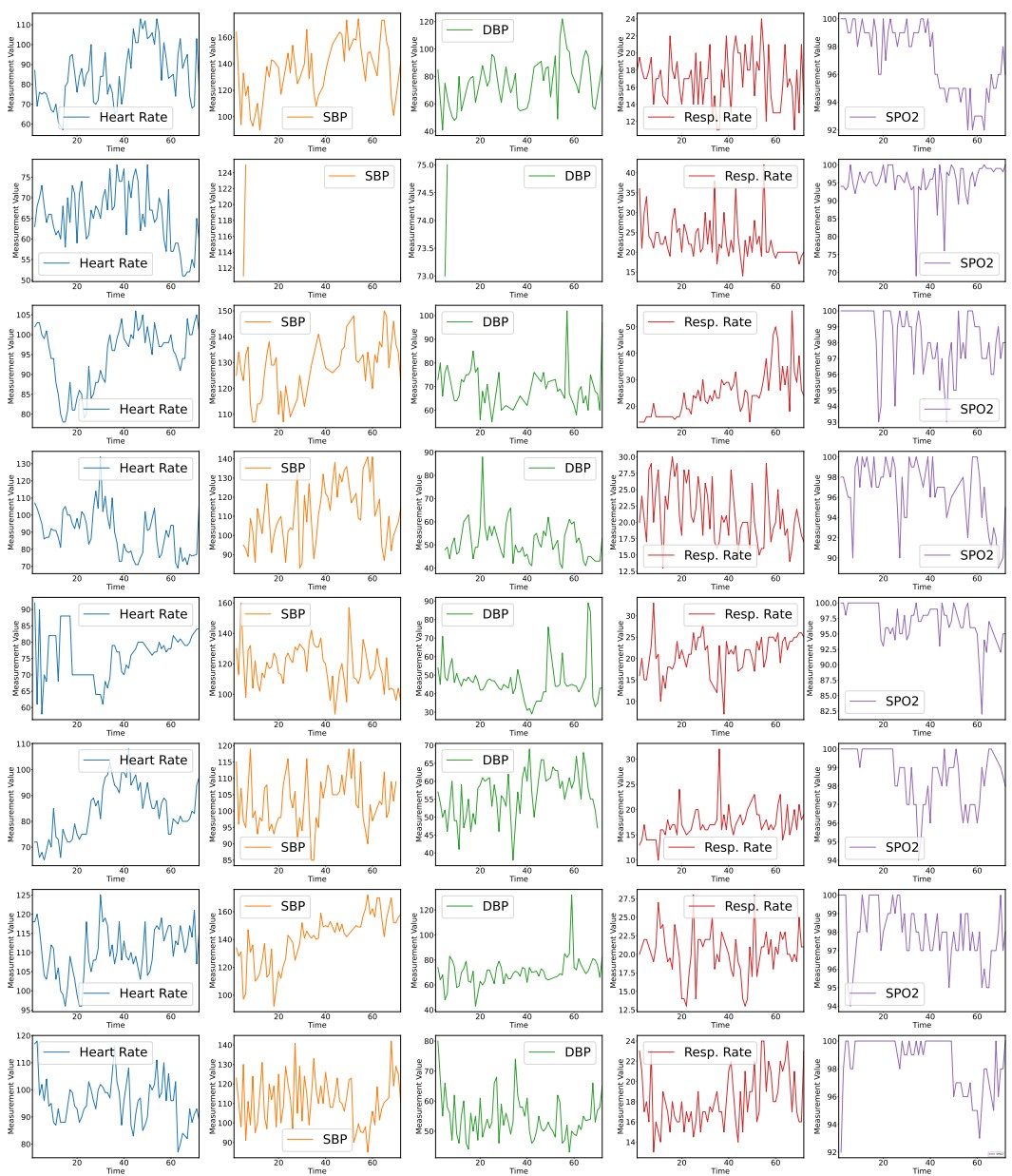

MIMIC-IV: time series in real testing data.

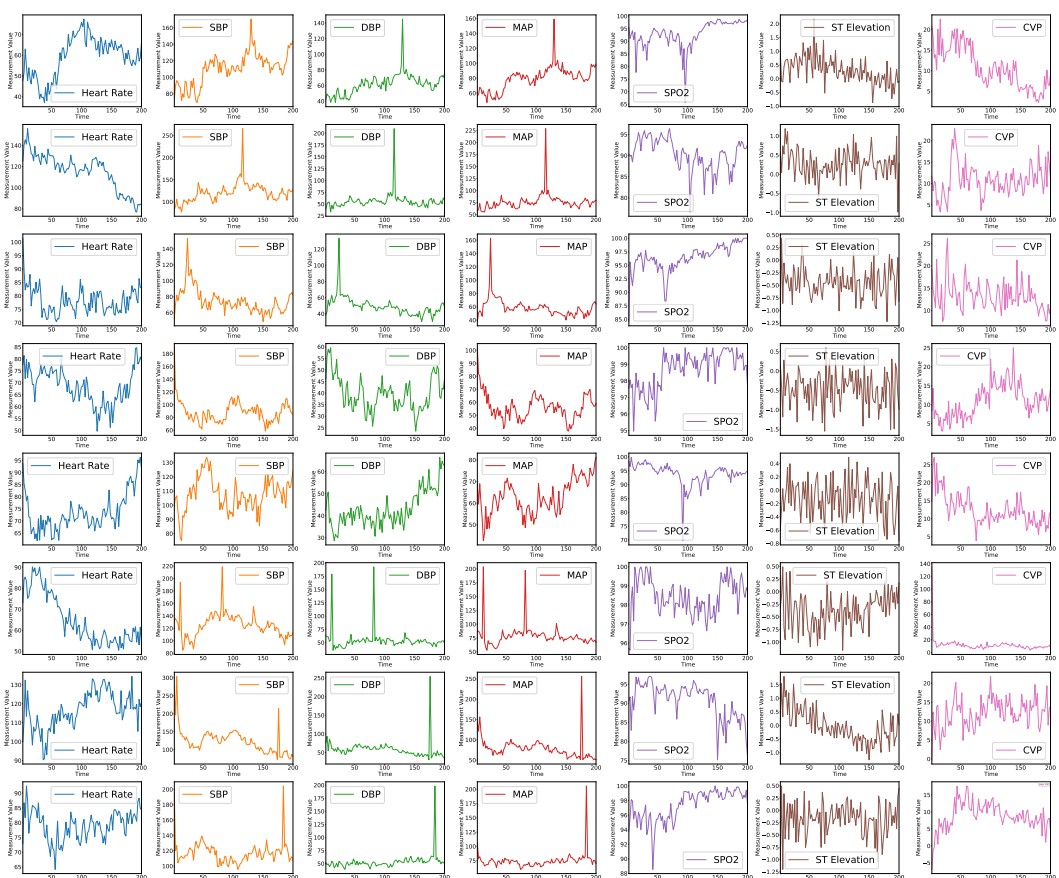

HiRID: synthetic time series produced by TIMEDIFF.

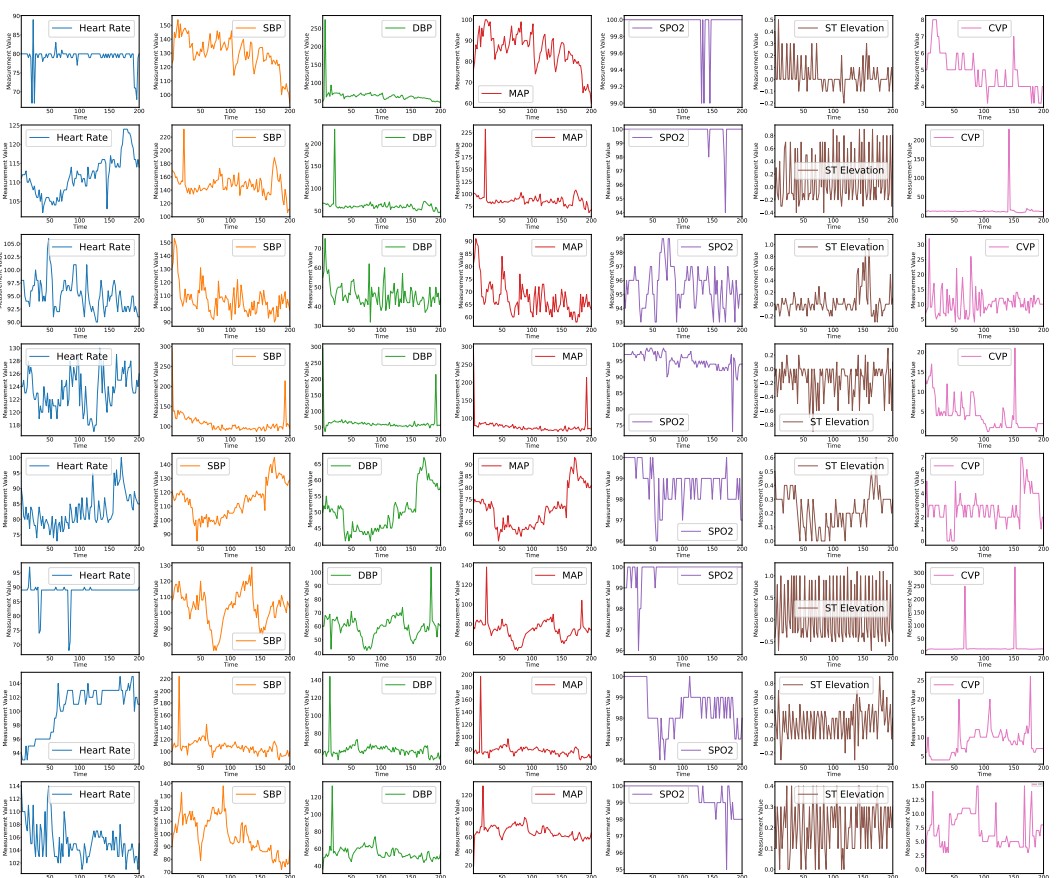

HiRID: time series in real testing data.

