# OpenReview forum: "Fast and Reliable Generation of EHR Time Series via Diffusion Models"
_ICLR.cc/2024/Conference — ICLR 2024 Conference Withdrawn Submission_

### Official Review · Reviewer_5r5j · 2023-10-31

**Soundness:** 3 good
**Presentation:** 3 good
**Contribution:** 2 fair
**Rating:** 3
**Confidence:** 4

**Summary:**

This paper developed a diffusion model, TimeDiff to generate synthetic EHR time-series data. Authors consider the generation of both numerical
(real-valued) and discrete time-series by combining both multinomial and Gaussian diffusions. Experiments on 6 datasets show the proposed method achieves better discriminative and predictive scores.

**Strengths:**

1. The paper studies an important problem, mixed-type EHR generation.

**Weaknesses:**

1. The authors claim that TIMEDIFF is the first to generate mixed type EHR. However, other works like [1,2,3,4] have done the same or similar things. The authors did not compare or discuss these works. And comparing one or two of them is important.
2. Further, the proposed method might not be as new or unique as the authors suggest. It’s important to note that TIMEDIFF is a diffusion model by replacing the U-Net architecture. The change of loss function and nosing step is straightforward by incorporating previous works (multinomial diffusion).
3. The authors say that TIMEDIFF is faster because it takes less time to train than GAN-based methods. However, when we look at how fast generative models work, we usually look at how quickly they can create samples (**sampling procedure**), not how quickly they can be trained. Diffusion models, which TIMEDIFF is based on, usually create high-quality samples but take a long time to do so. So, saying that TIMEDIFF is more efficient than GANs in the introduction is misleading. The authors should instead focus on comparing the speed of creating samples.
4 Privacy evaluation is necessary as existing works do, like membership inference attack.

## Reference
1. Li et.al., 2023. Generating synthetic mixed-type longitudinal electronic health records for artificial intelligent applications
2. Ceritli et.al. 2022.  Synthesizing Mixed-type Electronic Health Records using Diffusion Models
3. Naseer1 et.al., 2023. ScoEHR: Generating Synthetic Electronic Health Records using Continuous-time Diffusion Models
4. Theodorou et.al., 2023. Synthesize high-dimensional longitudinal electronic health records via hierarchical autoregressive language model

**Questions:**

See weakness.

---

> ### Author Response · Authors · 2023-11-17
> **Response to Reviewer 5r5j**
>
> We thank the reviewer for the comments. We address each of the questions and weaknesses below.
>
> > The authors claim that TIMEDIFF is the first to generate mixed type EHR. However, other works like [1,2,3,4] have done the same or similar things. The authors did not compare or discuss these works. And comparing one or two of them is important.
>
> First, we would like to kindly note that our work states that TimeDiff is the first to generate **mix-type time series via diffusion models. We did not claim our work is the first to generate mixed-type EHR**, which is already being studied by others. Our contribution comes from the use of mixed diffusion to generate mixed-type time series, which has not been done before.
>
> Secondly, we acknowledge the importance of comparing our work with the method in [1], EHR-M-GAN. In response, we have conducted an evaluation and included these results in our revised manuscript. This comparison clearly demonstrates TimeDiff's superiority over all other eleven baseline methods considered in our study. We hope our comparison is convincing.
>
> Thirdly, we cannot compare with [2,3,4] for the reasons below:
> * For [2]: this method is for **tabular** data generation, which is very different from **time series** data generation.
> * [3,4] became publicly available in August, 2023, while our submission deadline is in September, 2023. We are not required to compare with these works per ICLR policy.
>
> > Further, the proposed method might not be as new or unique as the authors suggest. It’s important to note that TIMEDIFF is a diffusion model by replacing the U-Net architecture. The change of loss function and nosing step is straightforward by incorporating previous works (multinomial diffusion).
>
> We appreciate your comments regarding the novelty of our method. While diffusion models or generative models for EHR are an established concept, their application to mixed-type time series data, especially in time-dependent Electronic Health Records (EHR), is a significant innovation in our work. This is not just a matter of substituting the U-Net architecture or altering the loss function. We have specifically tailored the noise model and loss function to address the complexities inherent in mixed-type EHR data, a challenge that existing models have not adequately tackled. This adaptation is crucial for accurately capturing the diverse data characteristics in EHR, as detailed in our paper. Ultimately, we believe our method enhances the accuracy and training efficiency of generative modeling in healthcare, presenting a noteworthy advancement.
>
> > The authors say that TIMEDIFF is faster because it takes less time to train than GAN-based methods. However, when we look at how fast generative models work, we usually look at how quickly they can create samples (sampling procedure), not how quickly they can be trained. Diffusion models, which TIMEDIFF is based on, usually create high-quality samples but take a long time to do so. So, saying that TIMEDIFF is more efficient than GANs in the introduction is misleading. The authors should instead focus on comparing the speed of creating samples.
>
> Our original intention for saying "Fast and Reliable Generation of EHR Time Series via Diffusion Models" was to indicate the ease in training. We have noticed that this can cause confusion for the readers. Thus, in our revision, we have removed "fast generation" in our title. We have also removed mentioning that our model is "more computationally efficient" in our introduction. Instead, we only mention that our method is easier to train than GAN-based methods. We hope these edits make our paper clearer to the readers.
>
>
> > Privacy evaluation is necessary as existing works do, like membership inference attack.
>
> We agree that this privacy risk should be evaluated. We have included membership inference risk in our revision. The results are shown in Table 4, and the full results are contained in Appendix B.4. Our method is able to achieve low risks across four EHR datasets.
>
> ## References
> [1] Li et.al., 2023. Generating synthetic mixed-type longitudinal electronic health records for artificial intelligent applications
>
> [2] Ceritli et.al. 2022. Synthesizing Mixed-type Electronic Health Records using Diffusion Models
>
> [3] Naseer1 et.al., 2023. ScoEHR: Generating Synthetic Electronic Health Records using Continuous-time Diffusion Models
>
> [4] Theodorou et.al., 2023. Synthesize high-dimensional longitudinal electronic health records via hierarchical autoregressive language model

---

### Official Review · Reviewer_27MX · 2023-10-31

**Soundness:** 2 fair
**Presentation:** 2 fair
**Contribution:** 3 good
**Rating:** 6
**Confidence:** 5

**Summary:**

The authors present an approach to generate synthetic EHR samples using denoting diffusion probabilistic models (DDPM). To admit both numerical and categorical data, they proposed a novel 2-stage method to generate samples using the diffusion model. Furthermore they compared against several baseline models for a number of tasks.

**Strengths:**

There are several key contributions in the paper as follows
- Synthetic samples for EHR is an immensely important topic that can potentially impact many aspects of AI for Health, including data availability and privacy preserving learning. The current SOTA method for synthetic EHR data is based of GAN. Seeing the promise of diffusion models for other domains, both in terms of performance and optimized training, it is thus quite exciting to see a working solution that can adapt to the nuances of EHR. The authors have explicitly considered several nuances such as mixture of numerical/categorical data and missing values.
- The performances on several benchmark datasets are quite promising, especially in terms of being able to mimic the real world datasets
- The authors have tried to justify the importance of several sub components using ablation studies

**Weaknesses:**

There are several aspects which if addressed can improve the exposition of the paper.
- The main aspect is that while the authors have performed various high level experimental evaluation, the paper is a bit under-analyzed, especially when considering the domain of healthcare. For example, it may be interesting to conduct sub-group analysis to understand reliability zones of the algorithm
- Another aspect that could be analyzed is some form of explainability analysis to understand the key driver of the learning. While the authors have presented results at a meta-level of categorical (multinomial) and numerical (gaussian) data modalities - it would be interesting to understand the modalities around health data dimensions such as diagnosis, drugs, and lab results.

Some other minor comments are as follows
- the presentation of the method can be substantially improved. While noting the page limit, the description of diffusion processes and the key contribution could be improved upon
- Some choices have not been explained in details. For example for the backbone network, the authors chose BiRNN. Were attention based models considered?
- Also, the baselines, while many, should include a few of the more recent architectures (e.g. based on diffusion processes that granted may not address the categorical data well) and some classical ones e.g MedGaN

**Questions:**

There are few aspects which may need some clarification from the authors
- the Diffusion process presented assumes no interaction between numerical and categorical features. Is this choice justified? Have the authors considered investigating individual trajectories for validity of the samples?
- The performance of in-hospital mortality task is rather low. Have the authors considered more advanced methods such as RNN/Attention models as modelers? In the same note, how was the data cohorted and the features selected for this task?

---

> ### Author Response · Authors · 2023-11-17
> **Response to Reviewer 27MX (first half)**
>
> We thank the reviewer for the thorough comments and thoughts. We provide our response for each of the points raised below.
>
> > The main aspect is that while the authors have performed various high level experimental evaluation, the paper is a bit under-analyzed, especially when considering the domain of healthcare. For example, it may be interesting to conduct sub-group analysis to understand reliability zones of the algorithm
>
> We agree with the reviewer and believe that sub-group analysis is very meaningful for a complete understanding of the proposed method. Unfortunately, due to time constraints in the rebuttal period, we cannot conduct a thorough sub-group and reliability zone analysis. We believe this is an interesting direction to explore nonetheless. We have noted this in Section 6 in our revision.
>
> > Another aspect that could be analyzed is some form of explainability analysis to understand the key driver of the learning. While the authors have presented results at a meta-level of categorical (multinomial) and numerical (gaussian) data modalities - it would be interesting to understand the modalities around health data dimensions such as diagnosis, drugs, and lab results.
>
> We agree that it would be interesting to evaluate the impact of different modalities of healthcare data on our model's performance. Although the current TimeDiff framework allows for the generation of categorical and numerical data modalities, our current work mainly focuses on generating non-sparse time series. For other modalities mentioned by the reviewer, they are often sparser than the data modalities we considered in this study. We hypothesize that a different modeling approach is needed since: (1) records like drugs and lab tests may not occur often and could be highly irregular; (2) the value for those results may be highly dependent on other conditions (such as initial diagnoses and past medical history). Thus, the complex nature of modeling such data modalities would require the design of a new approach, which is a good direction for future work. We have noted this and the need for explainability analysis in Section 6 as well.
>
>
> > the presentation of the method can be substantially improved. While noting the page limit, the description of diffusion processes and the key contribution could be improved upon
>
> We thank the reviewer for pointing out a potential improvement of our paper. We have edited the method and key contribution sections to make the presentation clearer in our new revision.
>
> > Some choices have not been explained in details. For example for the backbone network, the authors chose BiRNN. Were attention based models considered?
>
> We have previously considered and experimented with attention based models and found that they do not bring performance boosts. We have edited the texts to make this clearer in our revision.
>
> > Also, the baselines, while many, should include a few of the more recent architectures (e.g. based on diffusion processes that granted may not address the categorical data well) and some classical ones e.g MedGaN
>
> We agree with the reviewer that more recent architectures (especially EHR-focused ones) should be included. Our primary reason for the lack of EHR time series generation frameworks are three fold: (1) code for some recent works (Yoon at al., 2023; Kuobet al., 2023) are unavailable. (2) most of the other works are very recent, within four months away from ICLR submission date. Per ICLR policy, authors do not need to compare such very recent works. (3) Most of the classical works on EHR synthesis are not focused on time series generation, which is one of the primary motivations for our study.
>
> For the classical methods like MedGAN, they are designed for tabular or ICD code generation tasks, which is quite different from the time series generation task we are interested in our study. Thus, we believe direct comparison with MedGAN is unsuitable. Nevertheless, we totally agree that a EHR-based generative model should be added among the baselines. Thus, we have added EHR-M-GAN (Li et al., 2023). The results are contained in the revision of our paper.

---

> > ### Comment · Reviewer_27MX · 2023-11-22
> > **Thanks for the response**
> >
> > Thanks for the detailed response. Based on the response, I am happy with my original rating of the paper

---

> > > ### Author Response · Authors · 2023-11-23
> > > **Thank you for your feedback**
> > >
> > > Thanks for your prompt and helpful response. We promise we will update our paper based on your suggestions in the camera-ready version should our paper be accepted.

---

> ### Author Response · Authors · 2023-11-17
> **Response to Reviewer 27MX (second half)**
>
> > the Diffusion process presented assumes no interaction between numerical and categorical features. Is this choice justified? Have the authors considered investigating individual trajectories for validity of the samples
>
> For the forward processes (either multinomial or Gaussian diffusion), there is no interaction between numerical and categorical features. This is because the noises are added independently. This choice has been widely adopted in the diffusion model paradigm. The DSPD/CSPD baselines in our study proposed to modify this noise by using stochastic processes to model a distribution over continuous functions. However, as our results demonstrate, such modeling approach may be unsuitable for EHR time series measurements.
>
> While the forward processes in TimeDiff does not assume interactions between features, the reverse process does. This is achieved by the mixed loss. The neural network $s_{\theta}$ is trained for the reverse process, learning to generate new synthetic samples by gradually denoising the latent noise. This reverse process is performed in parallel for categorical and numerical features, which are both fed into $s_{\theta}$. Thus, the reverse process does capture interaction between numerical and categorical features, since the network must learn to approximate the reverse process via the mixed loss.
>
> We have investigated the individual trajectories from a statistical perspective, some examples are added to Appendix B.5 in our revision. For a medical-knowledge-oriented perspective, a thorough analysis should involve medical practitioners' judgements. Given the time constraint in the rebuttal period, we believe this is a great direction for future work instead.
>
>
> > The performance of in-hospital mortality task is rather low.
>
> Indeed, the performance for in-hospital mortality prediction task is low. We believe the major cause is the number of feature types used for training of the modelers. Since our approach solely focuses on time series features, the modelers cannot make use of static variables. These static variables can be very important for in-hospital mortality prediction tasks (an example is age and commorbidities). Without those static variables, it is difficult for the modelers to predict mortality alone by solely using time series.
>
> However, we would like to note that this is more of an issue with the prediction modelers rather than TimeDiff. The TSTR and TSRTR evaluations still demonstrate that the synthetic samples from TimeDiff can support modelers’ performances. This reflects the synthetic data's high utility, which is also supported by the discriminative and predictive scores in Table 1.
>
> > Have the authors considered more advanced methods such as RNN/Attention models as modelers?
>
> We have added the results for RNN modelers for in-hospital mortality prediction task in Appendix B.4.
>
> > In the same note, how was the data cohorted and the features selected for this task?
>
> Our feature selection and cohort selection methods for the real data are discussed in Appendix A.1. We have also made edits to Appendix A.1 to make the presentation clearer. To train the modelers, we first obtained synthetic samples from TimeDiff. We then used these synthetic samples as the training data for the modelers. We did not manually select the synthetic data samples.

---

### Official Review · Reviewer_WAfz · 2023-11-01

**Soundness:** 2 fair
**Presentation:** 2 fair
**Contribution:** 2 fair
**Rating:** 5
**Confidence:** 5

**Summary:**

The authors adopt DDPM (Gaussian transition together with multinomial transition) for EHR generation. They adopt Time-conditional BRNN as the backbone together with Diffusion Step Embedding. They evaluate their methods on six datasets against seven baseline methods.

**Strengths:**

The experimental results are good. The authors adopt six criteria rather than only TSTR and similarity criteria by previous methods.

**Weaknesses:**

The contribution of this paper is more heuristic, i.e., Time-conditional BRNN with time embedding can achieve better performance for generating EHR data while with no theoretical guarantees.

**Questions:**

The author adopts the sample mean as the imputation methods for dealing with missing data. More advance techniques can be adopted, which might further improve the performance.

---

> ### Author Response · Authors · 2023-11-17
> **Response to Reviewer WAfz**
>
> We thank the reviewer for the comments. We address the weakness and question raise by the reviewer below.
>
> > The contribution of this paper is more heuristic, i.e., Time-conditional BRNN with time embedding can achieve better performance for generating EHR data while with no theoretical guarantees.
>
> We believe theoretical understanding of diffusion or score-based models are very important to the ML community. However, **as indicated by our Primary Area of submission (applications to physical sciences), the focus of our paper is on *application* of diffusion models on EHR time series**. Most application-oriented studies in deep learning community are based on empirical analysis and experimentation, which we have conducted in our paper.
>
> We firmly believe theoretical explanations of the existing complex generative models are a crucial direction for future research. However, for this paper, our focus is on application rather than theory. We have added theoretical analysis as a future direction in Section 6.
>
> > The author adopts the sample mean as the imputation methods for dealing with missing data. More advance techniques can be adopted, which might further improve the performance.
>
> We thank the reviewer for pointing out this idea.
>
> Firstly, we have explored using spline interpolation to replace missing values, which is a more advanced technique than using the sample mean. We found it yields similar performances.
>
> Secondly, we adopt sample mean since it is straightforward to implement, allowing for easier comparisons between baseline methods. This allows our evaluation to focus on the generative models themselves rather than missing value imputations. Given our paper's focus on generative modeling, optimizing missing value imputations is relatively out of the scope of this work.

---

> > ### Comment · Reviewer_WAfz · 2023-11-23
> >
> > Thank you very much for the detailed responses. My overall rating remains.

---

### Official Review · Reviewer_UTQr · 2023-11-01

**Soundness:** 2 fair
**Presentation:** 3 good
**Contribution:** 2 fair
**Rating:** 3
**Confidence:** 5

**Summary:**

The paper proposed a diffusion probabilistic model for the generation of EHR time-series data, leveraging a combination of multinomial and Gaussian diffusion. By introducing this mixed diffusion approach specific to EHR time-series data, they have empirically demonstrated enhanced performance in comparison to other time-series generation methodologies, especially in terms of data utility.

**Strengths:**

- This is the first work to apply this mixed diffusion approach to EHR time-series data.
- The authors have demonstrated the model's performance not only on EHR data but also on non-EHR data, showcasing its applicability across diverse domains.

**Weaknesses:**

- The time series EHR synthesis studies from Kuo et al. (2023) and Yoon et al. (2023) were mentioned in the related works, but not included in the baseline section. It would be an imperative first step to improve the soundess of the paper to integrate these studies into the baseline to ensure a thorough comparative analysis, especially given the absence of synthetic models specifically designed for EHR synthesis in the current baseline candidates.
- The title suggests "Fast and reliable generation", yet the evaluation on this aspect seems somewhat limited. Merely measuring training time and claiming "fast generation" may be an overclaim, without addressing the sampling time.

**Questions:**

Please see the weaknesses

---

> ### Author Response · Authors · 2023-11-17
> **Response to Reviewer UTQr**
>
> We thank the reviewer for the critique on our paper. We address each of the weaknesses below:
>
> > The time series EHR synthesis studies from Kuo et al. (2023) and Yoon et al. (2023) were mentioned in the related works, but not included in the baseline section. It would be an imperative first step to improve the soundess of the paper to integrate these studies into the baseline to ensure a thorough comparative analysis, especially given the absence of synthetic models specifically designed for EHR synthesis in the current baseline candidates.
>
> The reviewer raised a great question about the soundness of our work. Our reasons for not including these two works in our paper are as follows:
> 1. We have considered to include these two baselines in our studies since the very beginning. **However, their code implementations are not avaialble. We also reached out to the authors for those two papers but received either no response or inability to share the code**. This is the main reason for not comparing TimeDiff with the works by Kuo et al. (2023) and Yoon et al. (2023).
> 2. In addition, the work by Yoon et al. is published in npj Digital Medicine on 11 August 2023 (which is after May 28, 2023). **This work falls under the ICLR policy where our submission can be excused from not comparing the results to it** (*see the last question in FAQ for Reviewers: https://iclr.cc/Conferences/2024/ReviewerGuide*):
>
> While we cannot obtain code implementation for the method proposed by Kuo et al., we would like to note that the alternative approach in our ablation study is very close to it. To summarize, the method used by Kuo et al. bascially applies Gaussian diffusion with a U-Net backbone, and discrete-valued time series is generated with argmax of softmax of the generated real-valued time series.
>
> We agree with the reviewer that most of our baseline candidates are not specifically designed for EHR time series generation (except for RCGAN). Thus, we have added EHR-M-GAN (Li et.al., 2023) as one of our baselines. We have included its results in the revision. Please see Table 1, Figure 1, Table 2, and Table 4 for the results. Due to page limits, we have to include some results in the appendix, and those can be found at Appendix B.1, B.2, and B.4.
>
> > The title suggests "Fast and reliable generation", yet the evaluation on this aspect seems somewhat limited. Merely measuring training time and claiming "fast generation" may be an overclaim, without addressing the sampling time
>
> We sincerely thank the reviewer for pointing out the confusion in our paper. Indeed, our original intention is to state that the training time is faster. As stated in our abstract, our work mainly focuses on demonstrating the superior performance of the diffusion model paradigm compared with generative adversarial networks in terms of ease of training and sample quality. We agree that this title can cause confusion for the readers. In our revision, we have adjusted the title and texts accordingly to clarify that we do not have speed up in sampling. We apologize for causing this confusion to the reviewer.

---

### Author Response · Authors · 2023-11-17
**Overview and Summary of Changes**

We thank the reviewers for their time and effort in providing insights and comments for our paper. We would like to provide some information on the commonly asked questions and weaknesses in our paper:
* **Lack of EHR generative models as baselines**: The reason for this is two-fold.
    1. Our work focuses on generating EHR time series. Most classical works such as medGAN, medBGAN, and medWGAN for EHR data generation are not designed for time series synthesis tasks.
    2. Most existing works on EHR time series generation are very recent, as a result, this either resulted in
        * (i) inability to obtain the code implementation even when we reached out to the authors (Kuo et al., 2023; Yoon et al., 2023)
        * (ii) published within four months of the ICLR submission deadline (Theodorou et al., 2023; Yoon et al., 2023; Naseer et al., 2023). Per ICLR policy, authors are not required to compare with very recent works like these (*see the last question in FAQ for Reviewers: https://iclr.cc/Conferences/2024/ReviewerGuide*)
        * (iii) incompatible setting since the work is on tabular data rather than on time series (Ceritli et al., 2022; Naseer et al., 2023).

Nonetheless, we have added a new baseline EHR-M-GAN (Li et al., 2023) since it is designed for the generation of mixed-type time series. The results are included in the revision. This new baseline is evaluated for all the metrics we considered in our study, along with the added metric of Membership Inference Risk (MIR). We found that TimeDiff outperforms EHR-M-GAN in terms of data utility metrics while being faster to train.

We also included new experiments to address the questions raised by some reviewers whenever possible.

Specifically, our newly added results are at: Table 1, Figure 1, Table 2, Table 4, Appendix B.1, Appendix B.2, Appendix B.4, and Appendix B.5.

* **Fast Sampling**: We acknowledge that our title, "Fast and Reliable Generation of EHR Time Series via Diffusion Models", could potentially imply the meaning that our work has fast sampling. We would like to kindly note that this is not our original intention. We intend to make the argument that TimeDiff is much easier and faster to train than GANs, which is also one of the merits of our proposed method. Thus, we have edited the title and introduction accordingly to avoid any potential source of confusion in our revision (this revision is only reflected in the PDF of our paper since we cannot change the title of our submission in OpenReview). Please note that this is a minor change by editing a few words, and the contributions of our paper still stay the same as our original submission.

### *References*
Li et al., 2023. Generating synthetic mixed-type longitudinal electronic health records for artificial intelligent applications

Ceritli et al. 2022. Synthesizing Mixed-type Electronic Health Records using Diffusion Models

Naseer et al., 2023. ScoEHR: Generating Synthetic Electronic Health Records using Continuous-time Diffusion Models

Theodorou et al., 2023. Synthesize high-dimensional longitudinal electronic health records via hierarchical autoregressive language model

Kuo et al., 2023.  Synthetic Health-related Longitudinal Data with Mixed-type Variables Generated using Diffusion Models

Yoon et al., 2023. EHR-Safe: generating high-fidelity and privacy-preserving synthetic electronic health records

---

### Author Response · Authors · 2023-11-21

Dear Reviewers,

We sincerely appreciate all your valuable comments. Since the deadline for the author-reviewer discussion period is approaching, we hope to check if you have any other concerns about our manuscript. Please do not hesitate to let us know if you need any clarification or have additional comments.

Best,
Authors